# Evaluating multiple candidates simultaneously reduces racial disparities in promotion and tenure

Theodore C. Masters-Waage[1], Juan M. Madera[2], Ebenezer Edema-Sillo [3], Ally St. Aubin [3], Peggy Lindner [4], Maritza Gaytan[5], Heyao Yu [6] ✉ & Christiane Spitzmueller [5] ✉

Black and Hispanic faculty – underrepresented minorities (URMs) within academia – face career barriers that come to a crux in promotion and tenure decisions. Leveraging a natural experiment in choice architecture within a dataset of 1804 promotion and tenure decisions across six universities, we find that joint (906 faculty) vs. separate (898 faculty) evaluation reduces racial disparities in faculty outcomes. Specifically, in joint evaluation, an analysis of the simple slopes finds that Black and Hispanic faculty receive, on average, 9% fewer negative votes at the department level than in separate evaluations when controlling for research productivity, school, gender, rank, discipline, department size, and grant acquisition. Using moderated mediation analyses, we calculate that this translates into a 16.2% increase in the likelihood of a Black/Hispanic faculty member receiving a promotion. In a survey of 289 professors who have served on promotion and tenure committees (i.e., the key P&T decision-makers), we find that only 17% of faculty expect joint evaluation to improve underrepresented minority faculty outcomes and, conversely, 43% expect separate evaluation to improve underrepresented minority faculty outcomes. This natural experiment suggests that altering evaluation mode or simulating joint evaluation mode could help address academia's under-representation problem, but not in the way decision-makers expect.

Black and Hispanic individuals comprise 31% of the US population but only 11% of faculty[1] and are thus defined as underrepresented minorities (URMs) within academia. This number decreases as you go up the ranks, with URM faculty more widely represented as Assistant Professors (13%) than in Associate (11%) or Full Professor ranks (8%). The low levels of URM representation in tenured faculty roles have been linked to racial disparities in the promotion and tenure process and to the notion of gatekeeping[2–4], where groups of high-powered decision-makers decide on the career progression of URMs. In academia, these gatekeepers are senior faculty who serve on promotion and tenure (P&T) committees and wield high power over junior faculty members' scientific careers and contributions. Although these gatekeeping decisions present as meritocratic, recent research has shown that - while accounting for scholarship, discipline, grant funding, and more - URM faculty receive a significantly lower proportion of positive votes relative to their non-URM (White or Asian) peers[2]. These racial disparities are damaging to science and, thus, society, as a more diverse academy is linked with increased innovation[5] and research

[1]Health Science Research Institute, University of California at Merced, Merced, CA, USA. [2]Conrad N. Hilton College of Global Hospitality Leadership, University of Houston, Houston, TX, USA. [3]Department of Psychology, University of Houston, Houston, TX, USA. [4]College of Engineering, Missouri University of Science and Technology, Rolla, MI, USA. [5]Office of the Vice Chancellor for Academic Affairs, University of California at Merced, Merced, CA, USA. [6]College of Health and Human Development, Pennsylvania State University, University Park, PA, USA. ✉e-mail: hvy5095@psu.edu; cspitzmueller@ucmerced.edu

productivity[6]. The goal of this paper is to examine how choice architecture could provide a pathway to addressing these disparities.

Policy makers and academics have noted the need for policy development and implementation evaluation to be grounded in rigorous empirical studies[7,8]. To accomplish this, there is a need for more experimental and quasi-experimental field research to test interventions that can help mitigate inequities. Moreover, these interventions need to be tailored to the context in which they are being employed. Existing policies to address racial disparities in organizations face a number of challenges in the context of promotion decisions. Some policies are grounded in research that suggests removing race from the equation by conducting a blind-evaluation process in which the race of the applicant is hidden from the evaluators[9]. These approaches appear to be supported by evidence but are not feasible in the majority of promotion cases where the decision-makers are already aware of the candidate's identity. Another policy has been to balance the systemic disadvantages faced by URM candidates through identity-conscious policies such as affirmative action, in which identity characteristics are acknowledged to reduce discrimination against URM candidates and promote equal educational and employment opportunities[10–13]. However, despite being effective in increasing representation, such approaches have been subjected to legal and ethical challenges - particularly in academia - and are challenging to implement and sustain. Another widely used group of policies seeks to broaden diversity among decision-makers, again a practice grounded in empirical evidence[14] but limited in application since it requires URM faculty to partake in significant institutional service. Yet other approaches aim to address the decision makers' perceptions of race through diversity or debiasing training[15]. Although such interventions have been implemented broadly across management, a recent meta-analysis of 492 studies on implicit bias training concluded that "changes in implicit measures are possible, but those changes do not necessarily translate into changes in explicit measures or behavior" (p. 1)[16]. In sum, although numerous approaches that could inform policy change and address racial disparities have been tried out in the real world, in academic settings, the effective approaches appear to be unfeasible, and the feasible approaches appear to be ineffective.

Through this paper, we propose to leverage choice architecture principles as a foundation for a novel set of policy recommendations. Such approaches have been effective in tackling gender bias in organizations[17,18], particularly when compared to the mixed effects of more frequently used approaches like diversity training[15]. Choice architecture refers to the design of decision-making environments, with nudges referring to when small contextual changes impact how choice options are evaluated. Leveraging choice architecture, in the form of nudges, has been discussed as an easy-to-implement policy intervention to address racial disparities in P&T; however, these ideas have yet to be tested empirically[19,20]. In this paper, we examined whether evaluation mode - if the URM promotion candidates were evaluated in isolation (separate evaluation) or simultaneously with other candidates seeking promotion (joint evaluation) - can mitigate racial disparities in the P&T process.

Racial minorities face disadvantages at multiple stages in their career. Partially, these disadvantages can be linked to persistent negative stereotypes about URMs in the workplace[21]. According to the theory of shifting standards[22,23], these negative stereotypes lead to double standards, where URM employees are judged to a higher standard compared to non-URM employees[2,22,24]. This can result in candidates with similar attributes being evaluated differently based on their race[22,23]; for example, for a URM employee, receiving a poor performance review one year may mean they do not get a promotion, but for a non-URM employee, they may be given the benefit of the doubt. This is exactly what has been observed in past research on P&T decisions in academia[2]. URM faculty are held to a higher standard when it comes to performance on traditional research metrics (h-index),

with non-URM faculty being given the benefit of the doubt for low performance, but the same standard is not being applied to URM faculty. The question this paper investigates is whether the negative stereotypes and double standards are amplified or mitigated in joint evaluation.

To do so, we draw from research on behavioral decision theory. This stream of literature has examined joint vs. separate evaluation modes in forced-choice paradigms - i.e., selecting between two options - predominantly in the domain of marketing[25]. However, the central hypothesis is very relevant to promotion decisions. According to this theory, in separate evaluations, individuals rely more on surface-level information - i.e., information that is easy to interpret - and less so on deep-level information[26,27]. To demonstrate this, consider that you are evaluating a female candidate on their leadership skills based on their past work performance. Without any comparisons to anchor performance standards (i.e., separate evaluation), it might be difficult to determine if this employee's performance is below average, average, or above average on leadership because evaluators lack a reference point. As such, evaluators are more likely to rely on surface-level information, such as a candidate's gender, and implicitly use gender-based stereotypes related to leadership[28]. Alternatively, in joint evaluation, evaluators do have a reference point because they can compare the two (or more) options to each other. This makes it easier to conceptualize the hard-to-evaluate information and be more deliberative in the evaluation process. This is what was found in the laboratory work by Bohnet, van Geen, and Bazerman[26]. Specifically, when predicting the future performance of a hypothetical employee, in a separate evaluation, decision-makers relied more on gender stereotypes (surface-level information), whereas in a joint evaluation, they relied more on past performance (deep-level information).

Through reducing the role of stereotypes in evaluations, we expect that joint evaluation will disrupt the use of double standards. A strong motivation in decision-making - and cognition more broadly - is to be logically consistent[29,30]. However, as is evidenced from double standards in the P&T process[2] and many other cases[22,23,31,32], decision-makers do not apply the same process consistently. Nevertheless, we argue that decision-makers are more likely to be consistent when evaluating two candidates jointly than when making these evaluations separately. This is because, in joint evaluation, the discrepancies in decision processes (i.e., the shifting standards[22]) will be more salient than in separate evaluation. Thus, from this theoretical perspective, we derive the hypothesis that joint evaluation will reduce racial disparities in promotion decisions.

That said, we also acknowledge the competing perspective that joint evaluation would worsen outcomes for URM faculty. This perspective is aligned with general social psychological theories of perception, that race is more salient when individuals are in a group (2 or more) than when they are alone[33,34]. For example, if a decision maker is evaluating a P&T candidate individually, their most prominent social feature might be that they are a faculty member; however, if they are evaluated with someone of a different race, it will become more salient that they are a Hispanic faculty member and that the other candidate is a White faculty member. Note, in a previous version of the manuscript, we formally hypothesized this competing perspective; however, on the suggestion of a reviewer, we decided to focus on a single hypothesis, which, based on the stronger theoretical rationale for the role of evaluation mode in the decision-making literature, was that joint evaluation would reduce racial disparities.

We examine this hypothesis in the context of P&T decisions. Due to the extremely low representation of other racial minority groups (N = 2; e.g., Native Alaskan, Pacific Islander, and Native American), we operationalized URMs as Black and Hispanic faculty and non-URMs as White and Asian faculty. The dataset includes a diverse set of six US public and private institutions, varying in size (~10,000 to ~80,000 students), from across four states, and ranging in their US

News national university ranking (rounded to the nearest ten universities rank as 50th, 50th, 130th, 170th, 170th, and 390th in the nation). Promotion policies can vary across institutions[35,36], however, in general, they follow a similar approach. Faculty are required to go up for promotion after a certain number of years (i.e., tenure clock), though they can go up early if they elect to, and they can receive an extension if their university permits (e.g., birth of a child). Once faculty have chosen to go up for promotion, most universities' faculty affairs units follow a rigid timeline determining, for example, when the candidate must submit their promotion portfolio and by when external review letters must be completed. Within this context, we define joint evaluation as when more than one candidate is being evaluated at the department-level within this allotted window. In other words, there is more than one P&T candidate going up for promotion to the same rank, in the same department, in the same promotion cycle. The exact timeline for P&T varied across the six institutions in our sample; however, they each allotted around one month for P&T decisions to be made at the department level (see Table S1). Therefore, when multiple candidates were evaluated in the same year, they will both need to be considered within this time window. Notably, this is a less rigid designation of joint evaluation compared to past laboratory studies where joint evaluation occurred at exactly the same time and was a forced-choice between two options[25], that said, the proximity of the evaluations still suggests that decision makers will use the candidates as a reference point for each other[26,37].

In this work, we investigate the effect of natural variation in evaluation mode (joint vs. separate evaluation) on 1804 P&T decisions made across the six institutions. Results from this natural experiment provide support for joint evaluation as a nudge to reduce racial disparities in promotion and tenure decisions. Moreover, being evaluated jointly with other promotion candidates significantly increased Uthe RM faculty's likelihood of receiving a promotion. These findings identify joint evaluation as a nudge for addressing racial disparities in promotion and tenure[2] and pave the way for future work to utilize choice architecture and nudges to improve validity in the academic promotion process.

## Results: Forecasting study

Prior to examining the role of evaluation mode in P&T decision making, we conducted a forecasting survey to examine gatekeepers' (i.e., professors') lay beliefs about how evaluation mode would influence racial disparities in the P&T process. After all, for policy to be implemented, it is important to know where key decision-makers stand on the issue to see if they need validation or convincing.

This project surveyed professors ($N = 285$) who had served on promotion and tenure committees before to determine whether their beliefs about evaluation modes' relationships with unfavorable evaluations of URM candidates were correct. A histogram of the results is presented in Fig. 1. As seen in the graph, the modal response was that the evaluation mode had "no effect". However, the data was skewed towards the left-hand side of the scale (i.e., that separate evaluation would reduce racial disparities), with 42.81% of participants predicting that separate evaluation would be "slightly" or "substantially" better; alternatively, only 17.19% predicted joint evaluation would be "slightly" or "substantially" better. In support of this, running a one-sample two-tailed Welch's t-test, we found that the mean of 2.56 (SD = 1.13; 95% CI [2.43, 2.69]) was significantly below the midpoint of 3, (t[284] = −6.59, $p < 0.001$, d = 0.78), supporting the conclusion that on average professors held the belief that separate evaluation would be better.

In sum, gatekeepers' lay beliefs are misaligned with our central Hypothesis with only a minority (17.19%) of participants expecting that joint evaluation would reduce racial disparities, and the vast majority of participants (82.81%) either believing that evaluation mode would have no effect or that separate evaluation would reduce racial disparities. It is particularly notable that the most common response from

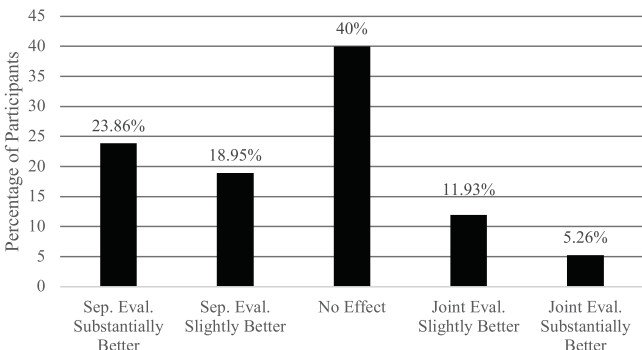

**Fig. 1 | Outcome from forecasting survey of faculty members ($N = 289$) with experience on promotion and tenure committees (i.e., gatekeepers).** Note. Percentages are provided as numbers at the top of each bar.

gatekeepers was in line with the competing hypothesis, that separate evaluation would reduce racial disparities. The contrast between this lay hypothesis and our initial hypothesis drawn from behavioral decision theory sets up the natural experiment as an empirical test to resolve these two contrasting perspectives.

## Results: Natural experiment
### No evidence of self selection

Given the high-stakes nature of promotion and tenure (P&T) decisions in academia, it is not feasible, or arguably ethical, to randomly assign candidates to either a joint or separate evaluation process. Therefore, we observed the effect of natural variation in evaluation mode across candidates to test the proposed competing hypotheses. We define this variation as quasi-random, meaning that the allocation to condition was not random but mirrors that of an experimental design, as the allocation to condition is unrelated to the dependent variable[38]. Quasi-experiments are powerful designs because they allow tests of causality in real-world settings[39-46]. In order to justify our classification of this as a quasi-experiment, we delve into what determines evaluation mode in the P&T process.

One threat to the integrity of a natural experiment is if candidates or decision-makers could self-select into an evaluation mode[44]. For example, if URM candidates believed there would be benefits of joint or separate evaluation and self-selected into one of these groups. However, given the structure of the P&T process, this is very unlikely. Within the university system, the time a candidate is granted before being evaluated for P&T is largely predetermined by university policy. For example, many universities in the US have a 6-year tenure clock, and by the end of this time period, faculty must go up for P&T. Decision makers in the P&T process have no control over this and thus cannot exert influence over a candidate's evaluation mode. Candidates also have limited influence over the length of these time periods. A notable exception is that candidates could ask for an extension to this period due to extenuating circumstances (e.g., the birth of a child) or decide to go up for tenure early[47]. However, based on the results of a two-tailed Welch's t-test, we see no evidence in our data that candidates were actively using these policies to influence evaluation mode, as individuals who were evaluated jointly vs. separately spent a similar amount of time in their previous rank (Joint: 6.06 years, Separate: 6.06 years; t(1744) = 0.02, $p = 0.983$, d = 0.001, 95% CI [−0.30, 0.31]) and took extensions at a similar rate (Joint: 7.06%, Separate: 7.35%; t(1800) = 0.23, $p = 0.815$, d = 0.01, 95% CI [−0.02, 0.03]). In fact, based on the results of two-tailed Welch's t-tests, faculty in joint and separate evaluation were very similar in terms of gender (t(1802) = 0.92, $p = 0.356$, d = 0.04, 95% CI [−0.02, 0.07]), URM status (t(1739) = 0.92, $p = 0.359$, d = 0.04, 95% CI [−0.02, 0.04]), h-index (t(1212) = −1.43, $p =$

0.152, d = 0.08, 95% CI [−0.19, 0.03]), university ranking (t(1800) = 1.32, *p* = 0.186, d = 0.06, 95% CI [−2.19, 0.11.25]), and grants (t(1578) = −0.85, *p* = 0.398, d = 0.04, 95% CI [−1.05, 0.42]); see Table 2. The only significant difference across faculty observed was in total votes, with candidates in joint evaluation having more department voters (t(1780) = −3.92, *p* < 0.001, d = 0.18, 95% CI [−2.45, −0.82]). However, mathematically, this is to be expected as it is more likely that a large department will jointly evaluate a candidate than a smaller department because there are more untenured faculty each year. Note, this variable (along with all the others mentioned) was included as a control. Moreover, evidence from the forecasting survey shows that the modal perspective from faculty is that the evaluation mode has no effect on P&T outcomes (40.13%). In sum, within the confines of the P&T system, self-selection from the decision makers or candidates into evaluation mode is challenging, and examining the one exception (candidates seeking extensions), we find no evidence of self-selection in our sample nor a clear expectation that one evaluation mode would improve their P&T outcomes.

Another threat to quasi-experiments is differences in environment between the two conditions[44]. For example, it is possible that joint or separate evaluation is more common at certain schools in our sample. To assess this, we examined the distribution of joint or separate evaluation cases across the six institutions in our sample. In the overall sample, the total number of candidates that were evaluated separately was 898 (49.78), and the total that were evaluated jointly was 906 (50.22%). Separated by institution, there are cases of joint and separate evaluation in all schools, with all schools having at least 43% of their decisions made in joint evaluation mode; see Table S2 for means and SDs of all study variables by institution. Further, separated by academic discipline, there are also cases of joint and separate evaluation in all disciplines with more than 3 cases. This distribution suggests that there were no major differences in the environment in which joint vs. separate evaluations took place. In addition, addressing another threat to the quasi-experimental design, the roughly even distribution of cases across institutions and disciplines means that there are no systematic differences in P&T policy across the conditions.

To conclude, based on the definition of Shadish and colleagues[38] and past quasi-experimental work by Talhelm and Dong[44], we classify the distribution of evaluation mode across P&T cases as quasi-random. As such, analyses will provide a causal test of the role of evaluation mode in P&T decisions[38–43,45,46,48]. Further, following the best practice recommendations for quasi-experiments[38,40,45,46,48], we strengthen internal validity by running a model including control variables to account for potential confounding variables[49]. These controls are detailed below, and justification for each variable is provided in the method section[35].

## Hypothesis tests

The central hypothesis of this paper is that joint evaluation will reduce racial disparities in voting outcomes for faculty, with the competing hypothesis derived from the forecasting survey that separate evaluation will reduce racial disparities. To do so, we use two operationalizations of voting outcomes at the department level. The first is the negative vote percentage (continuous variable), and the second is whether the vote was unanimous or not (binary variable). Results are reported for both sets of analyses below. Descriptive statistics are shown in Table 1, along with the subset demographics for candidates being evaluated jointly vs. separately (see Table 2) and URM vs. non-URM candidates (see Table 3).

**Negative vote percentage.** We regressed negative vote percentage on URM status, evaluation mode, and their interaction, with the interaction providing the critical test of the hypothesis. As shown in Table 4, analyses found evidence for an interaction effect (b = −0.10, SE = 0.02, *p* = 0.003, 95% CI [−0.16, −0.03]). Specifically, evident from Fig. 2, the

**Table 1 | Study 2 means, standard deviations, and correlations**

| | M | SD | 1 | 2 | 3 | 4 | 5 | 6 | 7 | 8 | 9 |
|---|---|---|---|---|---|---|---|---|---|---|---|
| 1. Joint evaluation | 0.50 | 0.50 | — | | | | | | | | |
| 2. URM status | 0.12 | 0.33 | −0.02(0.359) | — | | | | | | | |
| 3. Promotion rank | 0.48 | 0.50 | 0.00(0.882) | 0.07(0.004) | — | | | | | | |
| 4. Woman | 0.37 | 0.48 | −0.02(0.356) | 0.10(<0.001) | 0.10(<0.001) | — | | | | | |
| 5. Tenure in rank | 6.06 | 3.27 | −0.00(0.983) | 0.05(0.040) | 0.28(<0.001) | −0.03(0.205) | — | | | | |
| 6. H-index | 0.00 | 1.00 | 0.04(0.154) | −0.10(<0.001) | 0.30(<0.001) | −0.15(<0.001) | 0.00(0.914) | — | | | |
| 7. External grants | 5.00 | 7.54 | 0.02(0.399) | −0.09(<0.001) | 0.23(<0.001) | −0.12(<0.001) | 0.04(0.087) | 0.26(<0.001) | — | | |
| 8. Department negative vote % | 0.08 | 0.20 | 0.06(0.013) | 0.08(.001) | 0.09(<0.001) | −0.02(0.381) | 0.09(<0.001) | 0.04(0.087) | .01(.613) | — | |
| 9. Department unanimous vote | 0.76 | 0.42 | −0.08(0.001) | −0.02(0.326) | −0.08(0.002) | 0.03(0.221) | −0.06(0.028) | −0.06(0.043) | −0.08(0.003) | −.71(<.001) | — |
| 10. Total department votes | 9.54 | 8.89 | 0.09(<0.001) | −0.09(<0.001) | −0.22(<0.001) | −0.07(0.002) | −0.09(<0.001) | 0.06(0.044) | .01(.622) | −.06(.018) | −.22(<.001) |

This table represents means, standard deviations, and correlations for Study 2 variables. Values in parentheses are p-values. Joint evaluation is coded 1 for joint evaluation and 0 for single evaluation. URM status is coded 1 for candidates who are underrepresented minorities (Black/African American or Hispanic) and 0 for candidates who are White/Caucasian or Asian/Asian American. Promotion rank is coded 1 for promotion to full and 0 for promotion to associate. Woman is coded 1 for women candidates and 0 for men candidates. Tenure in rank refers to the number of years a candidate has been in their present rank. H-index refers to the candidate's h-index at the time of P&T. External grants refer to the number of external grants awarded as principal investigator. Department negative vote percentage is calculated as the total number of no votes divided by the total number of votes cast for a candidate. A department's unanimous vote is coded 1 for a unanimous vote and 0 for a non-unanimous vote. Total department votes are used as a proxy for department size.

**Table 2 | Breakdown of the demographics for separate vs joint evaluation and results of Welch's t-tests for comparison across conditions for each variable**

| | Women | | | H-Index | | | Extensions | | | Total Department Votes | | | Years in Present Rank | | | US News University Ranking | | | External Grants as PI | | | URM Status | | |
|---|---|---|---|---|---|---|---|---|---|---|---|---|---|---|---|---|---|---|---|---|---|---|---|---|
| Separate evaluation | 38.20% | | | 14.96 | | | 7.35% | | | 8.72 | | | 6.06 | | | 138.49 | | | 4.84 | | | 12.74% | | |
| Joint evaluation | 36.09% | | | 15.80 | | | 7.06% | | | 10.35 | | | 6.06 | | | 133.96 | | | 5.16 | | | 11.31% | | |
| Welch's t-test | t | | p | t | | p | t | | p | t | | p | t | | p | t | | p | t | | p | t | | p |
| | 0.92 | | .356 | −1.43 | | 0.152 | 0.23 | | 0.815 | −3.92 | | <0.001 | 0.02 | | 0.983 | 1.32 | | 0.186 | −0.85 | | 0.398 | 0.92 | | .359 |
| 95% CI | −0.02, 0.07 | | | −0.19, 0.03 | | | −0.02, 0.03 | | | −2.45, −0.82 | | | −0.30, 0.31 | | | −2.19, 11.25 | | | −1.05, 0.42 | | | −0.02, 0.04 | | |
| Cohen's d | 0.04 | | | 0.08 | | | 0.01 | | | 0.18 | | | 0.001 | | | 0.06 | | | 0.04 | | | 0.04 | | |

Two-tailed Welch's t-tests were conducted.

difference in negative votes between URM and non-URM candidates (i.e., racial disparities), was greater in separate than joint evaluation.

To further examine the effect on negative vote percentage, additional models were run in a stepwise manner to rule out alternative explanations[39]; note that due to missing data (see Table S3), these models also reduced sample size and, thus, statistical power. First, as shown in Table 4, a model ($N = 1535$) was run that incorporated candidate gender, school code, CIP Code, promotion rank, and years in present rank, which replicated the original finding (b = −0.09, SE = 0.03, $p$ = 0.005, 95% CI [−0.16, −0.03]). Second, as shown in Table 5, results also replicated when candidates' productivity (h-index and number of grants as PI) was included as control variables (N = 1027), addressing the possibility that candidates in joint evaluation differ in terms of scholarly ability (b = −0.13, SE = 0.04, $p$ = 0.003, 95% CI [−0.21, −0.04]). Third, as shown in Table 5, to address concerns that the observed effect may be due to differences in department size between joint and separate evaluation - e.g., joint evaluation and positive P&T outcomes are more likely in large departments - we controlled for the *number of total votes* as a proxy for department size ($N = 1535$) and replicated results (b = −0.09, SE = 0.03, $p$ = 0.005, 95% CI [−0.16, −0.03]). Finally, as shown in Table 5, we ran a model including all control variables in the same model ($N = 1027$) and again replicated the results (b = −0.13, SE = 0.04, $p$ = 0.004, 95% CI [−0.21, −0.04]).

To probe the shape of the interaction, a post hoc analysis of the simple slopes was conducted using the model with all the controls included (see Table S4). For URM faculty, going up for P&T jointly—as compared to separately—led to a 9% reduction in negative votes received at the department level ($t(983)$ = -2.23, b = −0.09, SE = 0.04, $p$ = 0.026, 95% CI [−0.17, −0.01]). For non-URM faculty, there was a significant relationship in the opposing direction ($t(983)$ = 2.62, b = 0.03, SE = 0.01, $p$ = 0.009, 95% CI [0.01, 0.06]). Consistent with our hypothesis, joint evaluation reduced racial disparities leading to a reduction in negative votes for URM faculty.

**Unanimous Votes.** Similar analyses were conducted to investigate the effect of evaluation mode on whether the department level vote was unanimously positive (1) or not (0). A model without control variables found no interaction between URM status and evaluation mode (Wald $\chi^2(1)$ = 1.13, OR = 1.08, $p$ = 0.289, 95% CI [0.94, 1.23]; see Table S5). The model including all the controls also did not find a significant interaction (Wald $\chi^2(1)$ = 1.88, OR = 1.14, $p$ = 0.171, 95% CI [0.95, 1.36]; see Table S5). Similar results were found using a generalized linear model (GLM) with a logit-link (Wald $\chi^2(1)$ = 0.04, OR = 0.39, $p$ = 0.084, 95% CI [0.13, 1.14]; see Table S6). In sum, although the pattern of results was in a consistent direction, the results do not replicate when using unanimous votes as the dependent variable.

**Provost Vote.** Next, we examined the downstream effect of evaluation mode at the department level on the provost vote. At the institutions in this dataset the Provost vote represents the ultimate decisions on whether a candidate will receive a promotion (1) or not (0). Therefore, to contextualize our result, we calculated the conditional indirect effect of URM status on provost vote via department-level negative vote percentage, depending on whether candidates were evaluated jointly or separately. In other words, how did the benefits of joint evaluation for URM faculty at the department level translate into likelihood of receiving a positive provost vote. Moderated mediation analyses found support for a conditional indirect effect (Index of moderated mediation = 0.61, $p$ = 0.025, 95% bootCI [0.15, 1.21]; see Table S7), meaning that the interaction between URM status and evaluation mode had downstream effects on provost vote via department level voting. We also examined the conditional main effect of URM status, i.e., regressing provost vote on the interaction between URM status and evaluation mode. The interaction term from these analyses was in the expected direction but was not statistically significant

**Table 3 | Breakdown of the demographics for URM vs. non-URM faculty**

| | Women | H-Index | Extensions | Total Department Votes | Years in Present Rank | US News University Ranking | External Grants as PI | URM Status |
|---|---|---|---|---|---|---|---|---|
| Non-URMs | 35.16% | 15.84 | 7.42% | 9.99 | 6.01 | 125.40 | 5.36 | 50.52% |
| URMs | 50.00% | 12.27 | 6.19% | 7.60 | 6.54 | 179.12 | 3.29 | 47.14% |

($b = -0.03$, SE = 0.04, $p = 0.460$, 95% CI [−0.12, 0.05]). Post-hoc tests found that, for a URM faculty, being evaluated jointly lead to a 16.2% increase in the likelihood of receiving a positive provost vote.

## Robustness checks

We conducted a series of analyses to ensure our results were robust to different model specifications. First, we conducted analyses with slight model modifications. We started by clustering the standard errors, replicating results when standard errors were clustered on university ($b = -0.13$, SE = 0.04, $p = 0.002$, 95% CI [−0.21, −0.05]) and department ($b = -0.13$, SE = 0.05, $p = 0.009$, 95% CI [−0.22, −0.03]; see Table S8). Testing another slight model modification, we ran a model with university-by-CIP fixed effects. We identified 103 University-by-CIP cells and ran the fixed effect finding results consistent with the OLS results ($b = -0.12$, SE = 0.04, $p = 0.037$, 95% CI [−0.23, −0.01]; see Table S9). Next, we ran a model relaxing our exclusion criteria. In the analysis plan we excluded Native American, Native Alaskan, Pacific Islander from analyses. Our initial rationale for not including these individuals was due to the small sample ($N = 2$) and the precedent set by other research by other teams using this dataset (e.g., Masters-Waage et al.[2]). As a robustness check we conducted analyses including these individuals and replicated our results ($b = -0.12$, SE = 0.04, $p = 0.004$, 95% CI [−0.21, −0.04]; see Table S10). Finally, due to the high amounts of missing data for h index and grants (see Table S3), we conducted analyses using a missing indicator approach by creating dummy variables for external grants and h index with missing values. Again, the interaction term between URM status and evaluation mode was significant ($b = -0.10$, SE = 0.03, $p = 0.003$, 95% CI [−0.16, −0.03]; see Table S11).

A second set of robustness checks focused on addressing model assumption violations. First, the data is highly-skewed with 75.5% of candidates receiving no negative votes at the department level. We also conducted a Breusch-Pagan test to examine whether the regression violated the homoscedasticity, and results showed that the data has a heteroscedasticity issue (BP index = 37.106, df = 9, $p < 0.001$). However, VIFs (ranging between 1.04 and 2.23) and Durbin-Watson analyses (DW = 1.97, $p = 0.324$) showed that the data were free from multicollinearity and autocorrelation issues. To statistically address both the skew and heteroscedasticity issues we use a 5000 times bootstrap resample method to test the robustness of the results. These results replicated support for our hypothesis ($b = -0.12$, SE = 0.03, $p = 0.004$, 95% bootCI [−0.20, −0.04]; see Table S12). In addition, to further address the issues related to skew, we ran a model including only candidates who received at least one negative vote (24.5% of the data). These analyses again replicated support for our hypothesis ($b = -0.30$, SE = 0.11, $p = 0.011$, 95% CI [−0.52, −0.07]; see Table S13), and increased the r-squared to 27%, which was higher than the model including candidates with no negative votes (r-squared = 9%).

A third set of robustness checks sought to account for the unique structure of the data. Decisions in our dataset were made at the department-level, with the vast majority of departments making multiple decisions across the seven-year window of data collection. In the primary analyses we controlled for department (i.e., discipline) and treated each of these decisions independently. However, an alternative approach would be to treat the data as a panel by grouping candidates by department and looking at decisions made over time. To conduct these analyses, we transposed the current dataset into a panel data with departments within each University as ID and year as the time variable. The panel data includes 326 unique departments in 7 years. The first set of analysis examined whether there was significant difference in joint evaluation across universities and departments. The results showed that there was no significant difference in joint evaluation across universities (F(5,1798) = 2.15, $p = 0.057$). The Tukey post-hoc analysis showed that none of the pairwise differences between schools are statistically significant at the 0.05 level. Following on from

**Table 4 | Stepwise OLS regression models for the interactive effect (OLS Regression) of URM status and joint evaluation (also termed Joint Eval.) on department negative vote percentage (Models 0-2)**

| Variable | Department Negative Vote % | | | | | | | | |
|---|---|---|---|---|---|---|---|---|---|
| | No Interaction | | | No Controls | | | Base Controls | | |
| | b | 95% CI | p | b | 95% CI | p | b | 95% CI | p |
| URM status | 0.05 | 0.02, 0.09 | 0.001 | 0.10 | 0.06, 0.14 | <0.001 | 0.09 | 0.04, 0.14 | <0.001 |
| Joint evaluation | 0.03 | 0.01, 0.05 | 0.012 | 0.04 | 0.01, 0.06 | 0.001 | 0.04 | 0.02, 0.06 | 0.001 |
| URM x Joint eval. | | | | −0.10 | −0.16, −0.03 | 0.003 | −0.09 | −0.16, −0.03 | 0.005 |
| Women | | | | | | | −0.003 | -0.03, 0.03 | 0.724 |
| Promotion rank | | | | | | | 0.02 | 0.00, 0.04 | 0.049 |
| Tenure in rank | | | | | | | 0.01 | 0.00, 0.01 | 0.001 |
| N | 1556 | | | 1556 | | | 1535 | | |
| $R^2$ | 0.02 | | | 0.01 | | | 0.07 | | |

OLS regression was conducted. Base controls included candidate (a) institution, (b) discipline, (c) gender, (d) promotion rank, and e) years in current rank. Institution and CIP Code were also used as controls in Model 2, but were not presented in the table due to the large number of parameters. URM status is coded 1 for candidates who are underrepresented minorities (Black/African American or Hispanic) and 0 for candidates who are White/Caucasian or Asian/Asian American. Joint evaluation is coded 1 for joint evaluation and 0 for single evaluation. Woman is coded 1 for women candidates and 0 for men candidates. Promotion rank is coded 1 for promotion to full and 0 for promotion to associate. Tenure in rank refers to the number of years a candidate has been in their present rank. H-index refers to the candidate's h-index at the time of P&T. External grants refer to the number of external grants awarded as principal investigator. Total dep. votes are used as a proxy for department size. No adjustments were made for multiple comparisons.

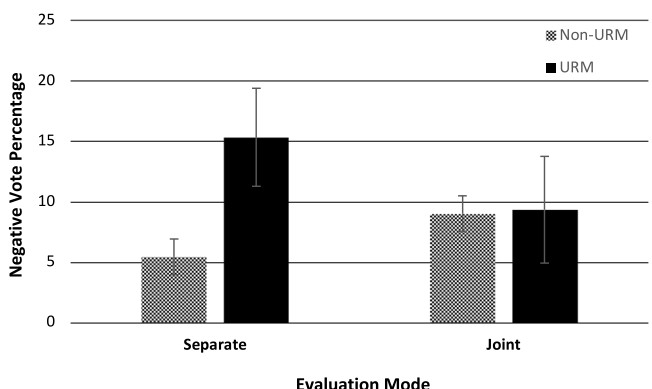

**Fig. 2 | Relationship between URM status and negative vote percentage at the department level based on regression estimates from the interaction effect in the base model (i.e., no control variables).** $N = 1556$ candidates. Data as presented represents the marginal means (accounting for covariates) for department negative vote percentage contingent on URM status and evaluation mode, therefore, the raw data (which does not account for covariates) is not displayed. However, the 95% confidence intervals visualize the variance for each mean value. URM refers to candidates who are underrepresented minorities. Non-URM refers to candidates who are not underrepresented minorities.

this, we replicated the analyses using a panel data approach. We first conducted a Hausman test to determine the appropriate analysis to perform. The results of the Hausman test suggest using the random effect model ($\chi 2 = 4.59$, $df = 3$, $p = 0.204$). The results from these analyses replicated support for our hypothesis in a model including all the control variables ($b = −0.12$, SE = 0.04, $p = 0.003$, 95% CI [−0.21, −0.04]) and across all models conducted in a stepwise manner (see Table S14).

**Supplemental Analyses**

We ran a series of supplemental analysis to test the parameters of our theory. First, our theory generally proposes that joint evaluation reduces racial disparities at both the promotion to associate and full professor level, hence our decision to group both types of decisions together in our analysis. However, to test whether the effect differs across decisions, we conducted subset analyses focusing on promotion to associate and full professor separately. Results replicated at the promotion to the associate professor level ($b = −0.13$, SE = 0.06, $p = 0.040$, 95% CI [−0.25, −0.01]; see Table S15) and the full professor level

($b = −0.15$, SE = 0.07, $p = 0.025$, 95% CI [−0.28, −0.02]; see Table S16), with a consistent interaction pattern between URM status and evaluation mode compared to the full-set analysis.

Another assumption of our theory was that the beneficial effect of evaluation mode for URM faculty would be consistent irrespective of the URM status of the faculty member they were being evaluated jointly with. To test this, we conducted analyses focusing on only cases in which joint evaluation occurred when URM candidates were being evaluated alongside non-URM candidates. Results from these analyses are in line with the conclusion above, finding that joint evaluation led to a lower negative vote percentage for URM faculty and an overall reduction in racial disparities ($b = −0.12$, SE = 0.05, $p = .006$, 95% CI [-0.21, −0.04]; see Table S17). Next, we sought to conduct a similar analysis focusing only on cases in which joint evaluation occurred when URM candidates were being evaluated alongside other URM candidates. Unfortunately the occurrence of this was very infrequent ($N = 28$), due to the underrepresentation of URM faculty in academia (roughly 8%) making it very unlikely two candidates would be evaluated simultaneously. Nevertheless we conducted the analyses and saw a similar pattern of results despite the interaction not being significant ($b = −0.14$, SE = 0.08, $p = 0.069$, 95% CI [−0.29, −0.01]; see Table S18).

A final supplemental analysis we conducted focused on the interaction between evaluation mode and gender (see Tables S19 and S20). Given that past research has found that evaluation mode can reduce biases experienced by women in the workplace[26], we sought to replicate the results reported above using gender as the moderator instead of URM status. Analyses revealed no evidence for an interaction between evaluation mode and gender in terms of negative votes (b = 0.03, SE = 0.03, $p = 0.284$, 95% CI [−0.02, 0.08]; see Table S19) nor unanimous votes (Wald $\chi^2(1) = 0.07$, OR = 0.99, $p = 0.793$, 95% CI [0.88–0.1.10]; see Table S20). However, it is also important to note that analyses of the main effect gender of voting outcomes found no evidence for gender disparities for P&T voting outcomes at the department level (negative vote percentage, $b = −0.01$, SE = 0.01, $p = 0.456$, 95% CI [−0.03, 0.01]; unanimous votes, Wald $\chi^2(1) = 1.11$, OR = 1.02, $p = 0.292$, 95% CI [0.98, 1.07]; see Tables S19 and S20). Therefore, the failure to replicate past work finding that evaluation mode reduces gender bias is likely due to the lack of evidence for a gender bias in P&T decision-making in the current dataset.

**General discussion**

Mechanisms to effectively address the continued underrepresentation of faculty of color in tenured positions in the academy continue to be

**Table 5 | Stepwise OLS regression models for the interactive effect (OLS Regression) of URM status and joint evaluation (also termed Joint Eval.) on department negative vote percentage (Models 3-5)**

| Variable | Department Negative Vote % | | | | | | | | |
|---|---|---|---|---|---|---|---|---|---|
| | Productivity | | | Department Size | | | All Controls | | |
| | *b* | 95% CI | *p* | *b* | 95% CI | *p* | *b* | 95% CI | *p* |
| URM status | 0.07 | 0.01, 0.14 | 0.024 | 0.09 | 0.04, 0.14 | <0.001 | 0.07 | 0.01, 0.14 | 0.023 |
| Joint evaluation | 0.03 | 0.01, 0.06 | 0.010 | 0.04 | 0.02, 0.06 | 0.001 | 0.03 | 0.01, 0.06 | 0.001 |
| URM x Joint eval. | −0.13 | −0.21, −0.04 | 0.003 | −0.09 | −0.16, −0.03 | 0.005 | −0.13 | −0.21, −0.04 | 0.004 |
| Women | −0.01 | −0.03, 0.02 | 0.730 | −0.004 | −0.03, 0.02 | 0.727 | −0.004 | −0.03, 0.02 | 0.739 |
| Promotion rank | 0.04 | 0.01, 0.07 | 0.006 | 0.02 | 0.00, 0.04 | 0.070 | 0.04 | 0.01, 0.06 | 0.016 |
| Tenure in rank | 0.01 | 0.00, 0.01 | 0.001 | 0.01 | 0.00, 0.01 | 0.001 | 0.01 | 0.00, 0.01 | 0.001 |
| H-index | 0.00 | −0.01, 0.01 | 0.946 | | | | 0.00 | −0.01, 0.02 | 0.885 |
| External grants | −0.001 | −0.00, 0.00 | 0.294 | | | | −0.001 | −0.00, 0.00 | 0.296 |
| Total dep. votes | | | | −0.00 | −0.00, 0.00 | 0.872 | −0.00 | −0.00, 0.00 | 0.621 |
| *N* | 1027 | | | 1535 | | | 1027 | | |
| *R²* | 0.08 | | | 0.07 | | | 0.08 | | |

OLS regression was conducted. Base controls included candidate a) institution, b) discipline, c) gender, d) promotion rank, and e) years in current rank. Institution and CIP Code were also used as controls in Models 3, 4, and 5 but were not presented in the table due to the large number of parameters. URM status is coded 1 for candidates who are underrepresented minorities (Black/African American or Hispanic) and 0 for candidates who are White/Caucasian or Asian/Asian American. Joint evaluation is coded 1 for joint evaluation and 0 for single evaluation. Woman is coded 1 for women candidates and 0 for men candidates. Promotion rank is coded 1 for promotion to full and 0 for promotion to associate. Tenure in rank refers to the number of years a candidate has been in their present rank. H-index refers to the candidate's h-index at the time of P&T. External grants refer to the number of external grants awarded as principal investigator. Total dep. votes are used as a proxy for department size. No adjustments were made for multiple comparisons.

scarce, with few data-informed interventions being available. The goal of this paper was to assess such an intervention, contrasting a joint vs. separate evaluation mode in P&T decision making[26,50]. In our analyses of a large quasi-experimental, consortium-based dataset, and controlling for a myriad of factors that influence decisions - including variation in research productivity, discipline, department size, and grant acquisition - joint evaluation was found to improve P&T outcomes for URM faculty. This opens the door to joint evaluation as an "evaluation nudge"[26] to address academia's underrepresentation problem.

This research makes a number of contributions to judgment and decision-making theory concerning the role of evaluation mode in applied settings. The setting of this research extends past work on joint vs. separate evaluation. First, this paper examined real-world (high-stakes) decision-making - as opposed to hypothetical lab decisions[26] or low-stakes product choices[25] - and demonstrated that joint evaluation is a feasible intervention in high-stakes contexts. Specifically, joint evaluation can reduce racial disparities by encouraging a comparison of candidate qualifications instead of defaulting to attending to candidate characteristics. Second, whereas past work has focused on individual decision making[25], this paper demonstrates that the benefits of joint evaluation are seen in a group decision-making situation. Related to the first point, this is important as in the real-world high-stakes decisions are often made in groups[51].

A third contribution to research on evaluation mode was that, unlike past work[25,26], decision makers were not in a forced-choice paradigm. In other words, instead of choosing between candidates, they were evaluating each candidate independently of the others. Therefore, the current findings demonstrate that the benefits of joint evaluation persist in broader contexts than previously thought. Even more pertinently, this allowed us to examine the effect of evaluation mode on URM and non-URM individuals separately. As observable in Fig. 2 and from the analysis of the simple slopes, along with joint evaluation improving outcomes for URM faculty, we also found that joint evaluation had a smaller but significant effect on outcomes for non-URM faculty in the opposite direction. This goes beyond past research by indicating that evaluation mode can also affect the majority group members, leading to harsher evaluations.

A fourth contribution concerns temporal proximity. Past work on evaluation mode has focused on contexts in which two options are compared simultaneously. However, in this research, that was not a requirement. First of all, although P&T decisions at each institution need to be made within a short time window (e.g., one month), they were not forced to be made in the same meeting. Further, even if they were conducted in the same meeting, the discussions of each candidate were likely to take place sequentially. On the one hand, the lack of temporal specificity of when these decisions occurred is a limitation of this dataset; however, it also provides a more general test of the benefits of joint evaluation, more akin to how these decisions are made in the real world. Therefore, finding that the benefits of joint evaluation replicate despite relaxing temporal proximity demonstrates the generalizability of the "evaluation nudge" theory.

This research also makes a number of contributions to policy within universities and beyond. From a university policy perspective, this paper points to a long-needed intervention for improving career progression outcomes for URM faculty. Although it is unfeasible for universities to force faculty to be evaluated jointly with others if they started in different years, there are methods that universities could apply to implement this intervention. First and foremost, in universities where P&T evaluations are required to be conducted in separate meetings (e.g., University of California system[52]), alleviating this policy would allow for more joint evaluation and its related benefits. Second, universities could engage in cluster hiring of URM and non-URM faculty to increase the likelihood of joint evaluation. Given that cluster hiring has already been shown to improve representation in the hiring process, this could be an effective strategy at two stages of URM faculty careers[53]. A third feasible (low cost) intervention we recommend is that P&T committees making decisions in a separate evaluation mode are provided with examples of past candidates to simulate a joint evaluation situation by supplying committees with a reference point when making their decisions[37]. Employing these approaches will help answer universities' and funding agencies' past calls for interventions to address the lack of representation of URM faculty in tenure-track and tenured faculty positions[54].

This research also identified the challenges policy makers may face when implementing these strategies due to a misalignment between decision makers' prior beliefs and the empirical evidence. As we identified in our survey, only 17% of P&T decision makers (i.e., gatekeepers) forecasted that joint evaluation would improve outcomes for URM faculty, whereas 43% forecasted that separate

evaluation would improve outcomes, and the remaining believed it would have no effect. This highlights that the "evaluation nudge" hypothesis was not intuitive to well-educated professors with experience making these decisions. Therefore, along with implementing structural policy changes, it is also important that policymakers look to update prior beliefs through education and training for P&T decision makers. Further, if professors have these prior beliefs, it is also possible that university policy makers also hold the same beliefs (e.g., Deans, Provosts, etc.), as many of these individuals previously held professor positions. This underlines the importance of education around the benefits of an "evaluation nudge".

Beyond universities, this paper also has practical implications for organizations more generally. As a case study, the university promotion system shares many similarities with other organizational systems, with promotion candidates being evaluated by a team of decision makers on a variety of key performance indicators. Moreover, within organizations, there is also variation in evaluation mode. For example, during annual or quarterly review,s employees are more-or-less jointly evaluated, as there is typically an entire cohort/department being evaluated within a given period. However, there are also cases where separate evaluations are very common, such as time-in-grade promotion decisions, which occur at varying intervals depending on employees' length of employment. Such decisions are common, including in the United States government, and can be mechanisms for systemic biases. For example, administrative data concerning the U.S. Patent Office, which uses time-in-grade promotions, demonstrates promotion delays and gaps for Black patent examiners[55]. Based on the results of this paper, further applied research should examine whether joint evaluation within the U.S. Patent Office could help reduce these racial disparities. The same also applies to the myriad of private organizations using separate evaluations that also benefit from a more top-down organizational structure than the higher education system[56], allowing them to implement a joint evaluation mode universally for promotion decisions.

The contributions of this paper should be considered in light of its strengths and limitations. Both a strength and a limitation of this paper is its use of a real-world dataset of high-stakes career decision-making in academic settings. The benefit of this approach is that it increases the ecological validity of the results[57]. Still, even though our study was conducted in a real-world setting with significant implications for society overall, the use of a single industry as a study context has limitations to the external validity of our findings. However, there are a few arguments that may assuage these concerns. First of all, the decision-making environment—i.e., individuals meeting to discuss the promotion of an employee and each voting—is similar to the promotion processes across many organizations. Second, the rigidity of the academic process, which follows clear institutional guidelines, means that this study could examine the evaluation mode in a structured manner. For example, investigating promotion decisions in another organizational domain where the protocol varies greatly would make it harder to isolate the effect of evaluation mode on decision-making.

Another challenge associated with the use of a real-world dataset is the complex inputs to decisions. As shown in Table 4, the model including all of the control variables accounts for 8% of the variance in negative vote percentage at the department level. An r-squared below 10% is not uncommon in applied psychological science[58,59], given the inherent complexity and noise in the psychological process[60]. Further, as shown in the robustness checks, the R-squared increases to 27% when addressing skew in the dataset by excluding participants with no negative votes. Nevertheless, it does highlight the challenges in predicting P&T voting behavior. This is notable given that measures of scholar productivity (h-index and grants obtained) are included in the model, and neither of them significantly predicted voting outcomes. Although past research has identified indirect effects of scholarly

productivity on voting outcome[2,61] the lack of a direct relationship demonstrates that P&T decision makers derive evaluations of research impact and quality through means other than citation metrics, such as relying more on their qualitative assessment of the candidates' contributions rather than on cumulative metrics[62]. Instead, it appears that P&T decision-making is a more complex process, highlighting the need for more research into this topic in order to identify the factors driving these high-stakes decisions.

There are also important future directions needed to expand this research. First, the low number of URM faculty in the dataset - particularly Native American, Native Alaskan, and Pacific Islander ($N = 2$) - limits the statistical power of these analyses. Therefore, despite the current data being the largest dataset of P&T decisions in the US, we encourage future high-powered research to replicate these results. Further, future studies should also seek to account for the role of more factors in the P&T process. Notably, coders at each institution did not record the racial demographics of the committee at the time of the P&T decision. This is partly due to the challenges of doing so, as committee members' identity is not always recorded in order to preserve confidentiality. Nevertheless, future research would benefit from examining how committee diversity is related to racial disparities in the P&T process.

To conclude, this paper provides support for an evaluation nudge to reduce racial disparities in P&T decisions. From a policy perspective, this is an approach institutions can explore as they strive to diversify their faculty to maximize student learning[63–65] and innovation[5,6]. However, this paper also points to the need to adjust decision makers' incorrect prior beliefs about evaluation mode through systematic training and education.

## METHOD
### Forecasting survey
**Participants.** This project surveyed professors who had served on promotion and tenure committees before to determine whether their beliefs about evaluation modes and unfavorable evaluations of URM candidates were correct. The sample was collected from across 11 universities in the United States. As a convenience sample, no statistical method was used to predetermine sample size, and no exclusions were used. A total of 285 participants completed the survey, of which 77 were women, with the following race/ethnic breakdown: White (72.60%), Asian (11.74%), Hispanic (7.12%), Black/African American (3.91%), Native American, Native Alaskan, Pacific Islander (.71%), or Other (3.91%).

**Ethics and Consent.** This study was approved by the University of Houston Institutional Review Board (IRB400004725) and complied with ethical regulations. Consent took place online prior to participants answering the online survey questions.

**Data collection.** Participants were recruited via their institution's email address. Participants were informed that previous research has identified racial disparities in the promotion and tenure process and were then asked to provide feedback on an intervention to address this disparity. Participants saw the following question prompt:

"Within the promotion and tenure process, for about half of P&T cases, there is a group of two or more candidates seeking a promotion to the same rank in the same department in the same year (instead of candidates being the only ones in a department seeking a promotion to that rank during that year).

*Do you think being evaluated as a group will lead to better P&T voting outcomes for URM faculty (right side of scale), do you think being evaluated individually will lead to better outcomes (left side of scale), or do you believe it doesn't matter? (middle of scale)"*

Participants then responded to this question on a 5-point scale ranging from "1" individual evaluation will be substantially better, to

"3" evaluation mode will have no effec,t to "5" joint evaluation will be substantially better.

## Natural experiment

**Participants.** This project uses data from the Center for Equity in Faculty Advancement (CEFA) database[66] which is a product of a collaborative consortium of ten American universities that are actively interested in studying promotion and tenure processes; used previously by Masters-Waage et al.[2]. The present study uses data gathered on P&T decisions for 1,804 P&T candidates from 2015–2022. Data was collected from across six research-intensive institutions (with the other four participating institutions not contributing data or data not being usable due to reporting formats) and included all available data; therefore, no statistical method was used to predetermine sample size. The dataset includes decisions to promote faculty to associate professor with tenure and full professor[2]. Most candidates were men (63%) and Caucasian/White (59%), and 48% of candidates were seeking Promotion to Full Professor.

**Consortium Information.** Our team built a consortium of universities to study promotion and tenure processes. In building this consortium, the goal was to bring together a diverse set of institutions in terms of size, research status, and ranking. In terms of student size, the consortium includes two smaller institutions (5000–15,000 students), three medium-sized institutions (40,000-50,000), and one larger institution (60,000+ students). The consortium primarily includes universities classified as Carnegie R1 institutions (i.e., very high research activity) but also one university classified as a Carnegie R2 institution (i.e., high research activity), providing variation of research activity. Further, whilst all institutions rank in the top 500 universities in the US according to the US News ranking, their position varies from 40th to 450th, with the median position being 157th. In addition, the set of universities includes two minority serving institutions, one designated as a Hispanic-Serving Institution (HSI) and another as a Historically Black College or University (HBCU). Finally, the six institutions span four US states providing geographical diversity.

All institutions were allotted approximately one month for P&T decisions to be made at the department level. Therefore, when multiple candidates were evaluated in the same year, they were both considered within this time window. For a full breakdown of the P&T timeline at each institution, see Table S1; note, the identity of the university was masked to preserve anonymity.

**Ethics and Consent.** This research complied with ethical regulations and, after review by the University of Houston Institutional Review Board (IRB), was declared to be non-human-subjects research (STUDY00002463; MOD00003374). IRB reliance agreements were then obtained from each consortium university. As described in the Data Collection section, each university's Academic Affairs leadership coded and then de-identified data that was shared with the leadership team. Consent was not obtained directly from participants, as de-identified archival data was used.

**Data Collection.** The data used in this study were collected to answer a host of research questions around P&T decision-making by the Center for Excellence in Faculty Advancement (CEFA). A subset of the analyses was previously reported by Masters-Waage et al.[2] (see paper for data collection details). Due to the sensitivity of the data collected, our team worked with members of the provost's office at each consortium partner institution to identify coders who already had access to confidential P&T data as part of their routine job responsibilities. These individuals, who followed a coding protocol developed by our team, coded institutional data relevant to the P&T process (e.g., voting outcomes) and candidate characteristics (e.g., discipline, race/ethnicity). To ensure the coding quality was high, we held bi-weekly meetings

with coders to answer questions. In addition, senior administrators at each partner university cross-checked 2% of cases for coding errors. Note, neither the coders nor the administrators were directly involved in the P&T decision-making process at their institution. Anonymized coding sheets were then merged into a dataset by the research team with an identifier used to scrape additional productivity data from online repositories (i.e., publication statistics, grants, and patents). The identifier used for scraping is stored in a database separate from the dataset used in the analyses, ensuring that the final dataset includes no identifiable information. For more details on the privacy steps taken in the coding process, see Masters-Waage et al.[2].

**Candidate URM status.** Candidate demographics were recorded in candidates' HR profiles (or an equivalent) and shared with us by the institution. Our partner institutions based their records on self-reported data that candidates provided (generally during the institution's onboarding process), meaning race/ethnicity was self-identified. A dummy variable was created indicating if participants were from an underrepresented minority (coded as "1") or not (coded as "0"), using the definition from the National Science Foundation[67]. The low number of individuals identifying as Native American/Native Hawaiian (0.4%; $N = 8$) and Other (2.7%; $N = 48$) makes it difficult to interpret the results for these ethnic groups; thus, they were not included in the analyses. Therefore, candidate URM status was coded as "1" for Black or Hispanic faculty and "0" for White or Asian faculty.

**Joint (vs. Separate) evaluation.** Joint evaluation is defined as when more than one P&T candidate is seeking promotion to the same role, in the same department, in the same academic discipline, and in the same year. A separate evaluation was any case where this was not occurring. The total number of candidates that were evaluated separately was 898 (49.78%), and the total number that were evaluated jointly was 906 (50.22%). The joint evaluation variable was converted into a dummy variable indicating "1" if the candidates were evaluated jointly and "0" if they were evaluated separately.

**Voting outcomes.** Promotion and tenure voting takes place within subsequent committee votes with department, college, and university-level committees, although some universities do not have separate college committees. The voting outcome variable used in this paper was committee voting at the department level (i.e., within the candidates' department). Although the dataset does include voting at the college and provost level, the joint evaluation occurred at the department level; for the college and provost level, it is likely that there are people being evaluated at the same time every year, given that they review candidates from a much larger pool.

As part of the voting process, committee members review candidate materials and convene to discuss and vote on P&T candidates, recording the number of votes for ("yes") or against ("no") a candidate, as well as the number of abstentions. From "yes" and "no" votes, two voting outcome variables were used for each of the six institutions investigated within the consortium: negative vote percentage and unanimous vote. Both these outcome variables are found to correlate with the provost vote, which represents the ultimate vote of whether a candidate receives a promotion or not.

**Negative voting percentage.** At the department level, negative voting percentage refers to the percentage of "no" votes that a faculty member received during each voting level. Overall, negative vote percentage is a more sensitive measure of committee support and allows for committee decisions to be compared across candidates beyond a binary variable about whether promotion was granted.

**Unanimous Votes.** A candidate was marked as having received a unanimous vote (1) if they received all "yes" votes. If a candidate

received any "no" votes, their voting outcome was marked as a non-unanimous vote (0). Abstentions were not factored into the coding of this dummy variable. Although unwarranted, unanimous votes are considered a gold standard to the extent that a non-unanimous decision can raise red flags when a candidate is reviewed by subsequent committees. Therefore, regardless of whether or not a promotion was ultimately granted, unanimous versus non-unanimous voting outcomes are a recognized distinction with real-world consequences.

**Control variables.** Given that the design of this study is quasi-random, the role of other confounding factors cannot be ruled out. Therefore, several control variables were included in the analyses to account for alternative explanations. As a baseline, analyses controlled for candidates' (a) institution, (b) discipline, (c) gender, (d) rank they are being promoted to (associate/full), and (e) years in current rank; note, all these variables were coded by individuals in the Provost office at each institution. In addition, other theoretically relevant variables, including scholarly productivity and department size, were included as controls.

**Scholarly productivity.** To account for the possibility that there are differing levels of scholarly productivity in candidates who go up for tenure separately as opposed to jointly, we controlled for two research metrics. The first was the h-index, which, despite being an imperfect measure of research impact, is used widely as a metric for judging academics in nearly all fields[68]; note, this variable was scraped from Google Scholar profiles at the time of the candidate's P&T decision. The second is the external grants, which refer to the number of external grants awarded as the principal investigator. This variable was coded by individuals in the Provost's office from candidates' CVs.

**Department size.** Another factor that is likely to vary across joint and separate evaluations is department size. This is because, mathematically, it is more likely that a large department will jointly evaluate a candidate than a smaller department because there are more untenured faculty each year. To calculate this variable, we used the total number of votes cast at the department level.

**Analysis plan.** All analyses were conducted separately for negative voting percentage and unanimous voting outcomes. Stepwise regressions were run to test hypothesis 1. Model 1 included only the key independent variables (URM status, joint evaluation, and their interaction term). Model 2 included the baseline controls (institution, discipline, gender, rank they are being promoted to [associate/full], and years in current rank). Model 3 included the baseline controls and scholarly productivity metrics (h-index and external grants). Model 4 included the baseline controls and department size (total department votes). Model 5 included all the control variables simultaneously. Additionally, we included a Model 0, in which we did not include the interaction term and only the two independent variables (URM status, joint evaluation). For analyses on negative vote percentage (continuous variable), OLS regression was used, reporting unstandardized coefficients. For unanimous votes (binary variable), logistic regression was used, reporting odds ratios.

**Reporting summary**
Further information on research design is available in the Nature Portfolio Reporting Summary linked to this article.

## Data availability
The forecasting survey and natural experiment data generated in this study have been deposited in the Open Science Framework (OSF) database accessible at: https://doi.org/10.17605/OSF.IO/JUX2C. The natural experiment uses the Center for Excellence in Faculty Advancement (CEFA) dataset[66], used previously by Masters-Waage et al.[2]. Due to the high sensitivity of this data, the raw natural experiment

is not available to protect the privacy of the promotion candidates in the dataset. However, a modified version of the dataset is available, omitting the US news ranking variable, as this would allow readers to identify each institution, and standardizing variables such as h-index to protect the privacy of the P&T candidates. This partial subset of the data is available at the OSF page.

## Code availability
The forecasting survey and natural experiment analysis code has been deposited in the Open Science Framework (OSF) database accessible at: https://doi.org/10.17605/OSF.IO/JUX2C.

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

## Acknowledgements

We gratefully acknowledge our grantors, including The Alfred P. Sloan Foundation and The National Science Foundation (NSF #1409928 (C.S.), NSF #2100034 (J.M.), and NSF #2411941 (C.S.)). Note that funders had no role in study design, data collection and analysis, decision to publish, or preparation of the manuscript. We would also like to thank our entire team within the promotion and tenure collaborative research consortium and our External Advisory Board members, who support the consortium.

## Author contributions

T.M.W., J.M.M, and C.S. are jointly responsible for the conceptualization of this study and funding acquisition. T.M.W. wrote the original draft, and T.M.W., J.M.M, M.G., A.G., E.E, H. Y., and C.S. jointly contributed to reviewing and editing the draft. T.M.W., E.E., P.L., and H. Y. are responsible for methodology, analyses, and data visualization. J.M.M. and C.S. supervised the project.

## Competing interests

The authors declare no competing interests.
