## [Transparent Peer Review file · Nature Communications]

Evaluating Multiple Candidates Simultaneously Reduces Racial Disparities in Promotion and Tenure

Corresponding Author: Professor Christiane Spitzmueller

Version 0:

Reviewer comments:

Reviewer #1

(Remarks to the Author)

The article is well written, incorporates a novel dataset, and covers an important topic. However, I do see some opportunities for improvement. They are described below in no particular order.

Put the sample size for the forecasting study in Table 1's notes.

Regarding Table 3, to determine if the key independent variables are indeed statistically significant, post-regression tests of joint significance should be performed (and disclosed) on the coefficients of URM, Joint, and URM x Joint.

Especially given that all 6 sampled institutions are research institutions, it is surprising that the research productivity variables (h-index & grants) are always statistically insignificant. While outside the scope of this study, this may be surprising enough to warrant increased investigation and discussion within the manuscript.

While not an overly important diagnostic, that the R^2 s are very small (0.09 or less) is somewhat noteworthy. The models appear to explain very little of the variation in promotion decisions. At least for the full model (the one with the full set of control variables), it would be interesting to see the partitioned R^2 s for at least the 3 key independent variables (URM, Joint, and URM x Joint). This would estimate an answer to how much of the variation is being explained specifically by these key variables. Given overall $R^2 = 0.09$, the answer will of course be "very little," but a quantitative estimate would be helpful.

I recommend adding this control variable: the percent of the department that is URM. Interactions with this new control may be warranted and enlightening.

The abstract states that 6 institutions were sampled. Then, on p.9, it says that 10 were. Finally, on p.18, it says that 6 institutions were chosen from a consortium of 10, clearing up the confusion. Still, the confusion is unnecessary.

The discussion of the supplemental analysis (joint evaluation cases where URM candidates were being evaluated alongside other URM candidates are excluded) is overly brief. These results should be presented in a new table. Results from the opposite subsample (only those cases included) should also be presented, perhaps in panel B of this new table. Further, it should be more clearly disclosed how this subsample was identified (Was it simply URM cases > 1 ?)

Descriptive statistics across URM status should be presented, as they were across Joint Evaluation status in Table 2.

Even if the results mirror one another, showing models across the subsamples of Assoc. Prof. promotions and Full Prof. promotions would be enlightening.

While it is perhaps unnecessary to tabulate all these additional coefficients, it would help the reader to know whether any other interaction terms with URM (e.g., h-index x URM) are statistically significant.

(Remarks on code availability)

n/a

Reviewer #2

(Remarks to the Author)

This paper presents a field test of the established finding that when candidates are evaluated jointly, this leads to more positive outcomes for populations that face negative stereotypes (e.g., women, URMs) than when candidates are evaluated separately. The test is conducted in a highly consequential setting: promotion and tenure decisions at universities. It involves the analysis of 1,804 such decisions at six universities and shows that when two or more candidates are under consideration for promotion or tenure by the same department in the same year, the positive vote share for minority candidates increases.

I have reviewed this paper (anonymously) previously for another journal, and it has been updated only slightly since my last review. Therefore, many of my comments will be similar to those the authors have received from me before. In addition to revisiting my previous points, I have also incorporated feedback from other reviewers that I believe would further enhance the paper, which I became privy to thanks to reviewing a prior draft of this paper for another journal . While I think the paper is already very strong, I have a number of suggestions that I hope will help the authors improve it.

COMMENTS ON LITERATURE REVIEW AND THEORY

(1) I found the inclusion of Hypothesis 2 on page 7 and the paragraph that preceded it a bit misguided. We have strong theoretical support and empirical support for Hypothesis 1 from decades of research on joint vs. separate decision making but absolutely no evidence for Hypothesis 2. I don't see a tension and suggest removing Hypothesis 2 and the short and unconvincing arguments presented to support it. Perhaps the logic articulated prior to the presentation of Hypothesis 2 could be mentioned when discussing the poor performance of lay forecasters who sought to predict the paper's findings? For non-experts who are unfamiliar with the large literature on joint versus separate decision making, perhaps the weak arguments made to support Hypothesis 2 came to mind. But the researchers writing this paper should focus exclusively on the well-established and proven theory behind Hypothesis 1 to justify their study.

(2) The paper would benefit from greater specificity and contextualization regarding its operationalization of "joint evaluation." While the authors define joint evaluation broadly—considering professors evaluated in the same calendar year as jointly evaluated—this deviates from traditional approaches that emphasize simultaneous assessments (i.e., within the same meeting). The analysis could be deepened by exploring whether temporal proximity of evaluations strengthens the "joint evaluation" effect; for example, are the effects more pronounced when evaluations occur in the same quarter, month, or meeting? If the authors do not have the ability to analyze this with their data, then I believe it is at least worth mentioning in the paper's discussion (and/or introduction) that an additional contribution of the paper is to show that it is not necessary for two cases to be evaluated simultaneously for the "joint evaluation" results from past studies to occur.

(3) I found it surprising that the paper only focuses on joint vs. separate P&T decisions for URM faculty and does not analyze joint vs. separate P&T decisions for women. The literature on joint vs. separate decisions has primarily focused on how women in stereotypically male domains benefit from joint evaluation. This should be acknowledged openly, and the authors should also run additional analyses with a focus on the promotion of women (instead of URMs) and report those results, too (even briefly). Whether an analysis of joint vs. separate evaluations of women faculty shows the same pattern or not, it would be interesting and informative to know this in light of past research (the analyses could be relegated to an appendix and mentioned briefly in the discussion, but they belong somewhere in the narrative).

(4) The paper would benefit from a more detailed discussion of the tenure process. For instance, they should define what constitutes a positive tenure vote (unanimous support? or merely a majority? And is this the same across institutions studied?). Moreover, the authors should offer a concrete interpretation of the observed 11% reduction in negative votes for URM candidates evaluated jointly. For instance, by making some broad assumptions about typical tenure and promotion policies, how might this reduction translate into an actual increase in the number of URM professors securing tenure or promotion? Addressing these points would significantly enhance the practical significance and clarity of the study's findings.

(5) I would also encourage the authors to explicitly position the contribution of the paper as extending beyond academia. The question of how evaluation mode affects outcomes by race likely applies beyond the specific context of university tenure committees to decision-making more broadly. Highlighting the generalizability of this phenomenon would significantly broaden the contribution of your paper.

COMMENTS ON EMPIRICS

(6) I appreciate the work that surely went into obtaining this remarkable, unique dataset. One challenge is that the distribution of negative votes is highly skewed. The median number of negative votes is zero in both conditions, and key result appears to be driven by a very small group of URM candidates who received a non-zero number of negative votes. Specifically, only 48 URM candidates received any negative votes. Assuming that it is not straightforward to obtain additional data from 2023 and 2024 to increase the sample size somewhat, I would encourage the authors to transparently note this limitation of the current analysis (that it hinges on these small number statistics.).

(7) Why not control for how many other candidates have been considered so far, in total, at the time of the candidate's vote? Or how far into the P&T season (e.g. in days/weeks) the candidate was considered?

(8) I found the justification for only looking at Black and Hispanic URMs and excluding Native Americans somewhat confusing. With an N of 2, why not include Native Americans at least in a robustness check? They are certainly URMs! The exclusion of the "Other" race category seems more justifiable since it is unclear what "Other" means.

(9) To enhance reproducibility and transparency, I strongly encourage the authors to upload all analysis scripts, materials (for the forecasting experiment), and more detailed documentation of variable creation on their OSF.

(Remarks on code availability)

Reviewer #3

(Remarks to the Author)

(Remarks on code availability)

Reviewer #4

(Remarks to the Author)

Comments on Joint Evaluation Reduces Racial Disparities in Promotion and Tenure: Evidence from a Natural Experiment

Summary:

The authors use variation in the "mode" of tenure voting, joint vs. single, across several institutions and departments using remarkable data on candidates' attributes combined with department-by-institution information on the voting process. They begin with a survey asking whether those gatekeepers think this would improve or worsen disparities in URM tenure cases. Framed within a cross-disciplinary literature on choice architecture the authors conclude that in fact URM candidates receive fewer negative votes when evaluated in the joint, as opposed to separate, case, contrary to what the gatekeepers thought would happen.

Comments:

I very much appreciate framing the paper in terms of choice architecture and anchoring – in fact I think there is a little room to expand on the role anchoring plays in this particular case. I appreciated the discussion here as it sets up the mechanisms for the experiment very nicely, this was excellent framing and I appreciated that you framed the potential for it to go either way. Starting with a survey of gatekeepers' beliefs enriched the study, in particular given the results.

I was unclear on how "treatment" is "assigned" here (using the terms lightly). You say in the back end of the paper that, "Joint evaluation is defined as when more than one P&T candidate is seeking promotion to the same role, in the same department, in the same academic discipline, and in the same year. Separate evaluation was any case when this was not occurring (p. 20)". How was the mode determined? Does it vary over time within department? Does it vary across departments within a university? I think yes to both of these but the extent of this is important for understanding where the variation is coming from. Page 11 partially discusses this, but this is an important aspect to be clear on. I took time to download and see the data (thanks for providing this, it is a great practice). I did not see a year indicator in the data, so it seems as if this is all running as a pooled analysis. If you have the data by year (I assume you don't or you would have included it) I suggest treating this as a panel.

Tables 1 and 2 could be expanded to provide a clearer picture of the populations. I suggest having table 1 as only a table of means (and SD's) of the universe of schools (or departments). I will assume that the pool is balanced (each school is in the data in each year). We might have statistics about the universities (size, category, possibly some measure of "rank" or competitiveness) separately by single vs. joint evaluations. I understand that there are 10 universities but later you say four universities had unusable data later on in the paper.

In the main model, you have institution and CIP code controls, meaning that you have what I believe are university and separately department (or field) fixed effects (from the table notes). This means that the variation could be coming from changes within university-departments over time, or is coming simply off of differences across departments within universities, possibly over time or not, it is unclear which, but is important. The data available for download show that there is within department variation, implying that either there were changes over time or the variation is simply random across years, most likely the former. One would have to assume that the switch across years was uncorrelated with any other changes that are related to the vote share, which is possible but unlikely. I might also suggest a first regression where the dependent variable is the treatment itself – joint eval, such that the reader can understand which universities/departments are more likely to feature this mode. Without a time element however, you can only learn so much from this.

I believe you're using a linear probability model (i.e. OLS) which is probably just fine, but because the dependent variable is a percentage (0-1) and is therefore bounded, using a generalized linear model (GLM) with a logit-link might improve precision. I also suggest clustering standard errors on universities (or departments), given the marginal results I think it's important to take inference seriously here. I tried this with the data and code provided and the standard errors shrink, which

is surprising, but shows there is more precision to be had. Likewise, given there is variation within university-CIP cells in joint evaluation you can actually have a university-by-CIP fixed effects and results are similar.

I'd also suggest putting the sample sizes in the results table. You note some drop out due to missing data – you could remedy that by setting the covariates for those observations (assuming they're not the dependent variable) to zero and including an additional dummy variable indicating that the control was missing to get some sample size back and rule out any issues from attrition across models. You lose about half the sample from the first to the last regression. Unless that attrition is random, it is worth figuring out which variables are causing the issue – it's difficult to determine how much the sample change affects inference. At the very least the reader should be aware of this.

As an even smaller note, I'd suggest making the dependent variable positive votes as opposed to negative, it's just easier to interpret the effect of joint evaluation as a "positive" effect. I apologize for what might seem like nit-picking here, please do not hear me saying I don't believe (or like) the results, the opposite is true – I want to make sure they are as strong, reliable, and clear as possible. Now on to actual comments.

The dependent variable has a pretty limited distribution. The data suggest that more than three-quarters of the votes were unanimous, meaning there is very little variation in the dependent variable to begin with. Possibly this explains the very small R², and why the unanimous variable outcome does not show effects. This is pretty important for the analysis, and for the reader.

I may be reading the table wrong, but my interpretation is that on net, URM candidates receive the same share of votes. The main effect (e.g. Model 1 of Table 3) has a positive coefficient of 0.10 for URMs and the interaction effect is -0.09. I take this to mean that in that model (1) in joint evaluations, URM's receive the same share of negative votes but when evaluated separately, they receive 9 percentage points fewer votes. I believe percentage points is the right interpretation here as the dependent variable is the percent of votes. I think you could add the baseline negative vote share to the table and then calculate what 0.09 is off of the base share of negative votes. On line 266 and in the abstract you say that the result is a 9% reduction in negative votes, which I believe is incorrect.

I was also struck by the fact that H-index and grants have no relationship to vote shares, and by the lack of negative effect for female candidates. Do these fit with other results from the literature? You might consider interacting these with URM (and with joint), but I am not clear on what the story to tell would be. To add, the R² is quite low considering you have CIP and university FE in the model (unless you reported the adjusted R², which is fine but just say so). It's hard to believe that all of those explain only a tiny fraction of the overall variance is explained by all of that information. Possibly this is due to the lack of variation in the dependent variable.

I think you can learn something from the vote share vs. the unanimous vote vis-a-vis your hypotheses, in particular the anchoring effect. That it is not statistically significant does not imply the result is not meaningful. Nonetheless, I think Figure 2 is worth highlighting more - non-URM votes are unaltered by the switch, but negative URM votes decline by two-thirds (10 percentage points off of a base of 15 percent)! You could do this for unanimous vs. not as well. I think this says something about the mechanism – opinions about non-URM's don't change, URM's are simply seen in a less negative light, which I think points more to the double-standard mechanism you mentioned in the framing section more than the negative stereotyping, though it is tough to pin the mechanism down.

(Remarks on code availability)

Version 1:

Reviewer comments:

Reviewer #1

(Remarks to the Author)

The authors have adequately addressed my concerns.

(Remarks on code availability)

Reviewer #2

(Remarks to the Author)

I thank the authors for their thoughtful and comprehensive response to the reviewers' comments. I appreciate the point-by-point responses provided in their rebuttal letter.

The revised manuscript shows improvements in several areas. Most notably, the authors have streamlined the theoretical framing by removing the competing Hypothesis 2. I also appreciate the detailed information about the tenure process, the expanded analyses (including gender effects and various robustness checks), and the broader positioning of the findings beyond academia.

I believe that the current version of this manuscript is very strong and I only have a few minor comments at this point

1. I appreciate the authors' effort to assess practical significance by linking evaluation mode to provost outcomes through negative vote share using a moderated mediation model. However, this approach restricts the analysis to a single pathway and assumes the effect of evaluation mode on the provost's decision operates only through department-level votes. While this is useful for isolating that channel, I recommend also presenting the total effect of evaluation mode on the provost decision (e.g., regressing the provost vote directly on the URM x Joint Evaluation interaction). This would provide a complementary and more policy-relevant estimate of the intervention's overall impact.

2. I did not find Figure 2 to be a terribly compelling visual display of the key interaction effect and might suggest considering other ways of presenting the effect visually that would be more intuitive to general audiences, perhaps with a simple bar chart. It would also be helpful to state in the Figure 2 notes which regression estimates were used to generate the image presented.

3. Ideally Table 2 would not only show summary statistics describing faculty and their departments by joint versus separate evaluation but would also present t-tests or two sample proportion tests to assess the balance of these summary statistics across conditions to provide statistical support for the claim that this was a valid quasi-experiment (reporting results in the section entitled "Quasi-Experimental Design" from page 9-12). An overall F-test could be used to assess balance given that some individual covariates are likely to be slightly imbalanced across the many tests conducted.

4. In my first read of the manuscript, I had not picked up on the fact that these votes took place at universities that are not, for the most part, terribly prestigious. I think providing this context in the abstract and introduction (that this analysis was done at primarily R1 universities ranked outside the top 100 in the United States) would be helpful. The results could plausibly differ at highly selective institutions, so being explicit about where this work was done up front seems important.

(Remarks on code availability)

Reviewer #3

(Remarks to the Author)

(Remarks on code availability)

Version 2:

Reviewer comments:

Reviewer #2

(Remarks to the Author)

I thank the authors for their responses to my comments, particularly regarding the practical significance analyses, the improved visualization in Figure 2, the statistical tests for balance across conditions, and the clarification of institutional context.

I believe the current version of the manuscript is very strong and makes an important contribution to the literature. I have no further comments.

(Remarks on code availability)

Reviewer #3

(Remarks to the Author)

(Remarks on code availability)

Reviewer #1 (Remarks to the Author):

The article is well written, incorporates a novel dataset, and covers an important topic. However, I do see some opportunities for improvement. They are described below in no particular order.

Authors' Response

We appreciate your words of support and the in-depth comments you have provided. Below, find a point-by-point response to each comment.

Put the sample size for the forecasting study in Table 1's notes.

Thank you for noting this. We have made this change - please see *Figure 1*, which is where we present the results of the forecasting study. Table 1 in the document is the means and SDs for the promotion and tenure study.

Regarding Table 3, to determine if the key independent variables are indeed statistically significant, post-regression tests of joint significance should be performed (and disclosed) on the coefficients of URM, Joint, and URM x Joint.

To address this comment we performed two types of post-regression tests, both aimed at strengthening the robustness of the IV, moderator, and the interaction term. We hope one or both of these analyses help address your query.

First, we conducted a test to determine the joint significance of both independent variables and their interaction term. Given our original model was OLS we used an F-Test to conduct this analysis and did so while including all control variables. The analysis found support for the joint significance of these three variables ($F(3,883) = 2.63, p = .049$). We now include these results in a footnote of the paper (see p13).

Second, we conducted a 5,000 times bootstrap resample method to check the robustness of the results for the model with all control variables in Table 3. The results are shown in Table S12 below and replicate the results of the paper. We mention these results on p15, in a new "Robustness Checks" section of the manuscript.

Table S12. Bootstrapped OLS regression results for the effect of joint evaluation on department negative vote percentage with 5,000 iterations.

Variable	Department Negative Vote %
----------	----------------------------

	b	SE	LLCI	ULCI
URM status	.07*	.03	.01	.13
Joint evaluation	.03**	.01	.01	.06
URM x Joint evaluation	-.12**	.03	-.21	-.04
Women	-.01	.02	-.03	.02
Tenure in rank	.001***	.002	.002	.01
Promotion rank	.03***	.01	.001	.06
H-index	-.00	.001	-.001	.001
External grants	-.001	.002	-.002	.001
Total department votes	-.001	.001	-.002	.001
R ²	.09			

Especially given that all 6 sampled institutions are research institutions, it is surprising that the research productivity variables (h-index & grants) are always statistically insignificant. Although outside the scope of this study, this may be surprising enough to warrant increased investigation and discussion within the manuscript.

Thank you for noting this; we agree it is an interesting point and now pick up on it in the discussion (p. X). We think this speaks to the intense complexity of P&T decisions, and the importance of a factor such as evaluation mode over specific metrics. Interestingly, past research has shown that factors like h-index may have an indirect effect on P&T outcomes via content of external review letters (Madera et al., 2024) and influencing perceptions of minority candidates (Masters-Waage et al., 2024), but it appears it is not a direct predictor of voting outcome. Nevertheless, given the relevance of research output in the P&T process, we still think it is appropriate to include these controls.

In the discussion we also highlight that the lack of direct relationship between scholarly productivity and voting outcomes is also interesting as it documents that faculty members evaluate scholarship to a large degree independent of scholarly productivity metrics,

potentially relying more on their qualitative assessment of the candidates' contributions than on cumulative metrics describing impact (citations, h-index).

For ease of reference, we provide the added section to the general discussion below (p. 22-23):

"...[M]easures of scholar productivity (h index and grants obtained) are included in the model and neither of them significantly predicted voting outcomes. Although past research has identified indirect effects of scholarly productivity on voting outcome^{2,62} the lack of a direct relationship demonstrates that P&T decision makers derive evaluations of research impact and quality through means other than citation metrics, such as relying more on their qualitative assessment of the candidates' contributions rather than on cumulative metrics⁶⁶. Instead, it appears that P&T decision making is a more complex process, highlighting the need for more research into this topic in order to identify the factors driving these high-stakes decisions."

While not an overly important diagnostic, that the R²s are very small (0.09 or less) is somewhat noteworthy. The models appear to explain very little of the variation in promotion decisions. At least for the full model (the one with the full set of control variables), it would be interesting to see the partitioned R²s for at least the 3 key independent variables (URM, Joint, and URM x Joint). This would estimate an answer to how much of the variation is being explained specifically by these key variables. Given overall R² = 0.09, the answer will of course be "very little," but a quantitative estimate would be helpful.

We agree that the R²s are noteworthy and believe there are both conceptual and practical reasons for this. First, while P&T decisions are sometimes presented as objective meritocratic decisions, in practice, these decisions are made anonymously without a clear rubric to follow. Therefore, in line with our previous comment, we would argue that it is very likely that a wide range of factors influence a P&T member's decisions, including idiosyncratic subjective assessments of a candidate. This suggests why objective metrics (e.g., h-index, grants, years in present rank, etc.) do not explain a large amount of the variance in P&T decisions.

In addition to this, it is also likely that the nature of the P&T voting data is also contributing to the R²s. There is a significant proportion of faculty who receive no negative votes (75.5%), meaning that there is no variance here for the model to predict. However, on suggestion of another reviewer, we ran a model excluding individuals who received zero negative votes. Using this subsample (see Table S13), we replicated our

results and found that the R^2 increases to 27%. Therefore, it appears that the skew in the data contributed to the low R^2 s.

Finally, we would also stress that in behavioural science an R^2 of below 10% is not surprisingly low. This is largely attributable to the high degree of noise observed in human decision making and human behaviour more generally. These challenges are also amplified when studying decision making in real-world environments. Therefore, when raising this point in the discussion we highlight this issue as a challenge as opposed to a limitation. We have included an added section to the discussion highlighting these issues and the supplementary table displaying the results for the subset analysis:

“Another challenge associated with the use of a real world dataset is the complex inputs to decisions. As shown in Table 3, the model including all of the control variables accounts for 8% of the variance in negative vote percentage at the department level. An r -squared below 10% is not uncommon in applied psychological science^{59,60}, given the inherent complexity and noise in the psychological process⁶¹. Further, as shown in the robustness checks, the r -squared increases to 27% when addressing skew in the dataset by excluding participants with no negative votes. Nevertheless, it does highlight the challenges in predicting P&T voting behavior.”

Table S13. OLS regression results for the interactive effect of URM status and joint evaluation on department negative vote percentage excluding candidates who received zero negative votes.

Variable	Department Negative Vote %	
	b	SE
URM status	.17*	.08
Joint evaluation	.03	.03
URM x Joint evaluation	-.30*	.11
Women	-.04	.03
Tenure in rank	.01*	.004
Promotion in rank	-.001	.04
H-index	-.001	.001
External grants	-.004	.002
Total department votes	-.01	.001
R^2	.27	

N = 375.

Finally, we also provide for you below the partitioned R²s (Table S21); note, to calculate these we used the method outlined by Lindeman, Merenda, and Gold (1980).

Table S21. Partitioned R for each variable in Table 3, model including all control variables.

Variable	Department Negative Vote %
	Partitioned R ²
URM status	.008
Joint evaluation	.008
URM x Joint evaluation	.02
Women	.0003
Tenure in rank	.02
Promotion in rank	.03
H-index	.008
External grants	.008
Total department votes	.003

I recommend adding this control variable: the percent of the department that is URM. Interactions with this new control may be warranted and enlightening.

Unfortunately, while we do control for discipline in our analyses, we do not have a variable indicating the percentage of the candidate's department who were URM, a variable not usually recorded in P&T files or archived in institutional records. The P&T data was coded by members in the provost's office at each institution using highly protected institutional data. Further, given that department composition changes over time and turnover particularly among URM faculty can be significant, this would require that URM representation was coded in each department for each year. Typically, universities do not have easily accessible data on department demographics separated by each year.

Therefore, while we would like to include this as a control variable, we do not think it is feasible given the nature of the dataset. In the revised manuscript, we highlight this limitation in the general discussion (p. 23).

The abstract states that 6 institutions were sampled. Then, on p.9, it says that 10 were. Finally, on p.18, it says that 6 institutions were chosen from a consortium of 10, clearing up the confusion. Still, the confusion is unnecessary.

Thank you for allowing us to clarify. Initially, the consortium that came together to create this dataset included 10 institutions. However, two institutions did not complete all required elements of the coding process and (after construction of the consortium) we learned that two of the smaller institutions had non-traditional ways of completing P&T voting, e.g., they did not clearly structure P&T committees or record votes. Therefore, these four institutions are not included in analyses as none of them have data on voting outcomes; the focal dependent variable. In the paper, to improve clarity, we now only refer to six institutions throughout and have included a footnote on page 9 explaining the lack of data for four institutions (this footnote is provided below):

“The Center for Excellence in Faculty Advancement (CEFA) dataset includes a total of ten institutions, however, only six institutions provided usable data for this paper. At the time of submission to the journal, two institutions had not completed the coding process, so we cannot include their data. Further, another two institutions had non-traditional ways of completing P&T voting - e.g., they did not clearly structure P&T committees or record votes - so we could not include this data in the analyses. In sum, these four institutions are not included in analyses as none of them have data on voting outcomes.”

The discussion of the supplemental analysis (joint evaluation cases where URM candidates were being evaluated alongside other URM candidates are excluded) is overly brief. These results should be presented in a new table. Results from the opposite subsample (only those cases included) should also be presented, perhaps in panel B of this new table. Further, it should be more clearly disclosed how this subsample was identified (Was it simply URM cases > 1?)

We appreciate the encouragement to further develop the supplemental analyses (p16-18). In the revised manuscript we now include supplemental analyses for a) the associate and full professor level separately, b) an interaction with gender, and c) when URM faculty are being evaluated compared to non-URM faculty vs. only URM faculty.

For the final supplementary analyses we have followed your guidance. First, we conducted analyses focusing on *only* cases where a URM faculty was compared to a non-URM faculty. These analyses replicate the findings from the primary analyses ($b = -.12$, $SE = .05$, $p = .006$; see Table S17). Second, we conducted analyses focusing on cases where URM faculty are *only* compared to URM faculty. Note, this is a rare case as the underrepresentation of URM faculty (8% across the US) makes it particularly unlikely that two or more URM faculty will be evaluated in the same P&T year; in our analysis there were 28 such cases before listwise deletions. Nevertheless, we find results that are a similar pattern to those seen across the paper though not reaching statistical significance

($b = -.14$, $SE = .08$, $p = .07$; see Table S18). These analyses are reported in the Supplementary Analyses section (p17)

Descriptive statistics across URM status should be presented, as they were across Joint Evaluation status in Table 2.

Great point, we now include that in the table (see Table 2).

Even if the results mirror one another, showing models across the subsamples of Assoc. Prof. promotions and Full Prof. promotions would be enlightening.

We have added these subsample results to the supplementary materials and mention them in the manuscript (see section title “Supplementary Analyses”, p16-17).

In a model including all of the control variables, find a similar pattern of results at both levels of analysis which is consistent with the overall finding, i.e., joint evaluation reduces racial disparities. Using the smaller subsamples, the result is statistically significant at the associate professor level ($b = -.13$, $SE = .06$, $p = .04$) and the full professor level ($b = -.15$, $SE = .07$, $p = .02$).

For ease of reference we include both of these supplementary tables below:

Table S15. OLS regression results for the interactive effect of URM status and joint evaluation on department negative vote percentage for candidates seeking promotion to associate professor.

Variable	Department Negative Vote %	
	b	SE
URM status	.10	.05
Joint evaluation	.01	.02
URM x Joint evaluation	-.13*	.06
Women	-.01	.02
Tenure in rank	.03***	.01
H-index	-.00	.001
External grants	-.003	.002
Total department votes	-.00	.001
R^2	.11	

Table S16. OLS regression results for the interactive effect of URM status and joint evaluation on department negative vote percentage for candidates seeking promotion to full professor.

Variable	Department Negative Vote %	
	b	SE
URM status	.07	.05
Joint evaluation	.06**	.02
URM x Joint evaluation	-.15*	.07
Women	-.01	.02
Tenure in rank	.004	.002
H-index	.001	.001
External grants	.00	.001
Total department votes	-.002	.002
R^2	.12	

While it is perhaps unnecessary to tabulate all these additional coefficients, it would help the reader to know whether any other interaction terms with URM (e.g., h-index x URM) are statistically significant.

Thank you for this suggestion and we agree that there are many interesting avenues to explore with this dataset. That said, we are hesitant to include these analyses for a couple reasons. First, we would like to keep this paper’s narrative focused on joint vs separate evaluation as a moderating factor. The interaction effect between URM and h-index is documented in our prior published work (2024) and published in Nature Human Behaviour.

In other words, we have focused this paper on, from a behavioral decision making perspective, an important phenomenon with significant policy implications: joint and separate evaluations. However, if there is another interaction you are interested in that concerns evaluation mode, we would be happy to test it and include it as a supplemental analysis.

Note, in the Supplementary Analyses section we now also model an interaction between gender and evaluation mode but the result is not significant.

Reviewer #1 (Remarks on code availability):

n/a

Reviewer #2 (Remarks to the Author):

This paper presents a field test of the established finding that when candidates are evaluated jointly, this leads to more positive outcomes for populations that face negative stereotypes (e.g., women, URMs) than when candidates are evaluated separately. The test is conducted in a highly consequential setting: promotion and tenure decisions at universities. It involves the analysis of 1,804 such decisions at six universities and shows that when two or more candidates are under consideration for promotion or tenure by the same department in the same year, the positive vote share for minority candidates increases.

I have reviewed this paper (anonymously) previously for another journal, and it has been updated only slightly since my last review. Therefore, many of my comments will be similar to those the authors have received from me before. In addition to revisiting my previous points, I have also incorporated feedback from other reviewers that I believe would further enhance the paper, which I became privy to thanks to reviewing a prior draft of this paper for another journal.

While I think the paper is already very strong, I have a number of suggestions that I hope will help the authors improve it.

Authors' Response

Thank you for your encouraging words and support for this project. Below please find a point-by-point response to each of your comments.

COMMENTS ON LITERATURE REVIEW AND THEORY

(1) I found the inclusion of Hypothesis 2 on page 7 and the paragraph that preceded it a bit misguided. We have strong theoretical support and empirical support for Hypothesis 1 from decades of research on joint vs. separate decision making but absolutely no evidence for Hypothesis 2. I don't see a tension and suggest removing Hypothesis 2 and the short and unconvincing arguments presented to support it. Perhaps the logic articulated prior to the presentation of Hypothesis 2 could be mentioned when discussing the poor performance of lay forecasters who sought to predict the paper's findings? For non-experts who are unfamiliar with the large literature on joint versus separate decision making, perhaps the weak arguments made to support Hypothesis 2 came to mind. But the researchers writing this paper should focus exclusively on the well-established and proven theory behind Hypothesis 1 to justify their study.

Thank you for raising this issue. The original motivation of us developing this study was past judgement and decision making (JDM) work identifying the benefits of joint evaluation, so hypothesis 1 was our central hypothesis. However, when discussing the project in early stages with social psychologists and colleagues in other disciplines, we found that many people predicted the opposite results, that separate evaluation improves outcomes for URMs. This was our motivation to include hypothesis 2 as a competing hypothesis and the associated theoretical arguments. However, we agree that these theoretical arguments are more convoluted, therefore, we appreciate the encouragement to remove hypothesis 2 and more tightly focus the paper on the JDM perspective.

That said, while we have removed hypothesis 2, we still maintain parts of the theorizing but have moved it to the discussion section for the forecasting study (p8-9). As you note, the forecasting survey shows that a significant number of faculty intuitively side with this hypothesis, so we think it is important to discuss why individuals may have this lay belief.

Overall, we believe this improves the flow of the manuscript and keeps the paper more tightly focused on evaluation mode as a mechanism. With this more localized focus we have also been able to streamline the introduction.

(2) The paper would benefit from greater specificity and contextualization regarding its operationalization of “joint evaluation.” While the authors define joint evaluation broadly—considering professors evaluated in the same calendar year as jointly evaluated—this deviates from traditional approaches that emphasize simultaneous assessments (i.e., within the same meeting). The analysis could be deepened by exploring whether temporal proximity of evaluations strengthens the “joint evaluation” effect; for example, are the effects more pronounced when evaluations occur in the same quarter, month, or meeting? If the authors do not have the ability to analyze this with their data, then I believe it is at least worth mentioning in the paper’s discussion (and/or introduction) that an additional contribution of the paper is to show that it is not necessary for two cases to be evaluated simultaneously for the “joint evaluation” results from past studies to occur.

We agree this is an important point. We have included in the methods a new section describing how we operationalize and distinguish this from past research on this construct.

We do not have precise data on the temporal proximity of evaluations. However, the majority of institutions (including all of the ones in our dataset) have very tight windows for P&T decisions to be made, meaning that decisions are typically made in close proximity to one another. In the revised submission we include Table S1, which details

the P&T timeline at the six institutions in this dataset and shows that in each case departments committees must complete P&T voting roughly within a one-month period. Therefore, when multiple candidates were evaluated in the same year they will both need to be considered within this time window. Therefore, critically, when making decisions in the joint evaluation condition it is likely that P&T committees were mentally aware of and considering both candidates simultaneously. We now highlight this in a footnote in the introduction (p7), provide more details in the methods sections (p26), and include the new table for reference (Table S1).

In addition, in the review manuscript we also identify the lack of temporal specificity as a limitation and an interesting contribution to research on joint and separate evaluations in the broader judgment and decision making literature. As you suggest, the slight temporal distance highlights that the benefits of joint evaluation are not dependent on simultaneous evaluation as an additional contribution of this research (p19). For ease of reference we provide this section below:

“A fourth contribution concerns temporal proximity. Past work on evaluation mode has focused on contexts in which two options are compared simultaneously. However, in this research that was not a requirement. First of all, although P&T decisions at each institution need to be made within a short time window (e.g., one month), they were not forced to be made in the same meeting. Further, even if they were conducted in the same meeting the discussions of each candidate were likely to take place sequentially. On the one hand, the lack of temporal specificity of when these decisions occurred is a limitation of this dataset, however, it also provides a more general test of the benefits of joint evaluation more akin to how these decisions are made in the real world. Therefore, finding that the benefits of joint evaluation replicate despite relaxing temporal proximity demonstrates the generalizability of the “evaluation nudge” theory.”

(3) I found it surprising that the paper only focuses on joint vs. separate P&T decisions for URM faculty and does not analyze joint vs. separate P&T decisions for women. The literature on joint vs. separate decisions has primarily focused on how women in stereotypically male domains benefit from joint evaluation. This should be acknowledged openly, and the authors should also run additional analyses with a focus on the promotion of women (instead of URMs) and report those results, too (even briefly). Whether an analysis of joint vs. separate evaluations of women faculty shows the same pattern or not, it would be interesting and informative to know this in light of past research (the analyses could be relegated to an appendix and mentioned briefly in the discussion, but they belong somewhere in the narrative).

Thank you for raising this important point. Although the focus of our work is on URM faculty, we examined gender in more detail based on your suggestion. Interestingly, we find no significant interaction between gender and evaluation mode in terms of negative votes ($b = .03$, $SE = .03$, $p = .28$) nor unanimous votes ($OR = 0.99$, 95% CI [.88-.1.10]). A reason for this could be that unlike for URM faculty, in our dataset, gender is not associated with differential voting outcomes. Therefore, the lack of gender difference in terms of voting outcomes means that there is no gender inequity for evaluation mode to mitigate. As you suggest, we mention this in the supplemental analyses for the paper. For ease of reference we provide this section below (p17-18):

“A final supplemental analysis we conducted focused on the interaction between evaluation mode and gender. Given that past research has found that evaluation mode can reduce biases experienced by women in the workplace²⁷, we sought to replicate the results reported above using gender as the moderator instead of URM status. Analyses revealed no evidence for an interaction between evaluation mode and gender in terms of negative votes ($b = .03$, $SE = .03$, $p = .28$; see Table S19) nor unanimous votes ($OR = 0.99$, 95% CI [.88-.1.10]; see Table S20). However, it is also important to note that analyses of the main effect gender of voting outcomes found no evidence for gender disparities for P&T voting outcomes at the department level (negative vote percentage, $b = -.01$, $SE = .01$, $p = .46$; unanimous votes, $OR = 1.02$, 95% CI [.98-1.07]). Therefore, the failure to replicate past work finding that evaluation mode reduces gender bias, is likely due to the lack of evidence for a gender bias in P&T decision making in the current dataset.”

(4) The paper would benefit from a more detailed discussion of the tenure process. For instance, they should define what constitutes a positive tenure vote (unanimous support? or merely a majority? And is this the same across institutions studied?). Moreover, the authors should offer a concrete interpretation of the observed 11% reduction in negative votes for URM candidates evaluated jointly. For instance, by making some broad assumptions about typical tenure and promotion policies, how might this reduction translate into an actual increase in the number of URM professors securing tenure or promotion? Addressing these points would significantly enhance the practical significance and clarity of the study’s findings.

We have taken this advice on board in the revision of the paper, making changes throughout the manuscript.

First, in the introduction we provide a clearer description of the tenure process, pasted below for ease of reference (see p7):

“We examine this hypothesis in the context of P&T decisions made within US institutions. Promotion policies can vary across institutions^{34,35}, however, in general they follow a similar approach. Faculty are required to go up for promotion after a certain number of years (i.e., tenure clock), though they can go up early if they elect to and they can receive an extension if their university permits (e.g., birth of a child). Once faculty have chosen to go up for promotion, most universities’ faculty affairs units follow a rigid timeline determining, for example, when the candidate must submit their promotion portfolio and by when external review letters must be completed. The particular part of the P&T process we focus on is the meeting of the P&T committee at the department-level, to make their decision on the candidate. Department P&T committee decisions are influential since department P&T committee members are direct colleagues and usually in the same or adjacent disciplines as the P&T candidate.

In this paper, we define joint evaluation as when more than one candidate is being evaluated within this allotted window. In other words, there is more than one P&T candidate going up for promotion to the same rank, in the same department, in the same promotion cycle. Notably, this is a less rigid designation of joint evaluation compared to past laboratory studies where joint evaluation occurred at exactly the same time and was a forced-choice between two options²⁶, that said, the proximity of the evaluations still suggests that decision makers will use the candidates as a reference point for each other^{27,36}.”

Second, in the methods, we provide a longer and more precise description of the tenure voting process.

“Promotion and tenure voting takes place within subsequent committee votes with department, college, and university level committees, although some universities do not have separate college committees. The voting outcome variable used in this paper was committee voting at the department level (i.e., within the candidates’ department). Although the dataset does include voting at the college and provost level, the joint evaluation occurred at the department level; for the college and provost level, it is likely that there are people being evaluated at the same time every year, given they review candidates from a much larger pool.”

As part of the voting process, committee members review candidate materials and convene to discuss and vote on P&T candidates, recording the number of votes for (“yes”) or against (“no”) a candidate, as well as the number of abstentions. From “yes” and “no” votes, two voting outcome variables were used for each of the six institutions investigated within the consortium: negative vote percentage and unanimous vote.”

Third, we liked your suggestion to conceptualize how an the shift in negative votes at the department level could affect the overall tenure process. To do so, we examined the downstream effect of evaluation mode at the department level on the provost vote. At the institutions in this dataset the Provost vote represents the ultimate decision on whether a candidate will receive a promotion (1) or not (0). Therefore, to contextualize our result, we calculated the conditional indirect effect of URM status on provost vote via department-level negative vote perception, depending on whether candidates were evaluated jointly or separately. In other words, how did benefits of joint evaluation for URM faculty at the department level translate into likelihood of receiving a positive provost vote. Moderated mediation analyses found support for a conditional indirect effect (Index of moderated mediation = .65, 95% bootCI = [.18, 1.25]); see Table S7), meaning that the interaction between URM status and evaluation mode had downstream effects on provost vote via department level voting. Post-hoc tests found that, for a URM faculty, being evaluated jointly lead to a 16.2% increase in the likelihood of receiving a positive provost vote. We provide these analyses in the results section (p14).

Table S7. Moderated mediation results for condition indirect effect of URM status and joint evaluation on Provost vote via department negative vote percentage.

Variable	Department Negative Vote %	Provost vote
	b	b
URM status	.08*	.66
Joint evaluation	.04**	
URM x Joint evaluation	-.13**	
Women	-.01	-.26
Tenure in rank	.003*	.38
H-index	-.000	.001
External grants	-.0002	-.01
Total department votes	-.0003	.02
Department Negative Vote %		-5.00***
R ²	.05	.36
	Effect	Bootstrap CI
Separate evaluation	-.39	(-.94, .06)
Joint evaluation	.26**	(.06, .49)
Index of moderated mediation	.65	(.18, 1.25)

(5) I would also encourage the authors to explicitly position the contribution of the paper as extending beyond academia. The question of how evaluation mode affects outcomes by race likely applies beyond the specific context of university tenure committees to decision-

making more broadly. Highlighting the generalizability of this phenomenon would significantly broaden the contribution of your paper.

Thank you for encouraging us to broaden the discussion of our paper, we have followed this advice and think the discussion section is greatly improved. In particular, we focus on the experiences of URM employees in other professional occupations which have a comparably high-stakes promotion system.

In addition, we also expand on the contribution of this finding to research on group decision making in the discussion (p18-19). As we mention in the manuscript, the majority of work on evaluation mode has focused on individual decision making. Finding that the benefits extend to group decision making broadens the potential effect that evaluation mode could have on organizational processes.

While we make these additions, we have also tried to stay close to Nature Communication's requirements that authors do not "go beyond the findings of the data", therefore, we present these expansions as a direction for future research as opposed to a direct implication of our study.

For ease of reference the relevant section of the discussion is pasted below (p21):

"Beyond universities, this paper also has practical implications to organizations more generally. As a case study, the university promotion system shares many similarities with other organizational systems, with promotion candidates being evaluated by a team of decision makers on a variety of key performance indicators. Moreover, within organizations, there is also variation in evaluation mode. For example, during annual or quarterly reviews employees are more-or-less jointly evaluated, as there is typically an entire cohort/department being evaluated within a given period. However, there are also cases where separate evaluations are very common, such as time-in-grade promotion decisions which occur at varying intervals depending on employees' length of employment. Such decisions are common, including in the United States government, and can be mechanisms for systemic biases. For example, administrative data concerning the U.S. Patent Office, which uses time-in-grade promotions, demonstrates promotion delays and gaps for Black patent examiners⁵⁴. Based on the results of this paper, further applied research should examine whether joint evaluation within the U.S. Patent Office could help reduce these racial disparities. The same also applies to the myriad of private organizations using separate evaluations, that also benefit from a more top-down organizational structure than the higher education system⁵⁵, allowing them to implement joint evaluation mode universally for promotion decisions."

COMMENTS ON EMPIRICS

(6) I appreciate the work that surely went into obtaining this remarkable, unique dataset. One challenge is that the distribution of negative votes is highly skewed. The median number of negative votes is zero in both conditions, and key result appears to be driven by a very small group of URM candidates who received a non-zero number of negative votes. Specifically, only 48 URM candidates received any negative votes. Assuming that it is not straightforward to obtain additional data from 2023 and 2024 to increase the sample size somewhat, I would encourage the authors to transparently note this limitation of the current analysis (that it hinges on these small number statistics.).

Thank you for raising this important point. We have added a section to the limitations section in the discussion where we highlight this (p23). Even in our data (which to our knowledge is the largest dataset of P&T decisions obtained), the low number of URM candidates is an unavoidable feature of studying racial disparities in academia, where URM faculty make up less than 10% of our sample. Notably, in this paper, the central hypothesis focuses on the interaction between URM status and evaluation mode. Therefore, we are not simply theorizing the positive effect of joint evaluation on URM faculty outcomes, we are also theorizing that this effect will not be present for non-URM faculty as majority group members, which we also find support for.

In addition, to address the skew in distribution of negative votes, we conducted robustness check with (subsample removing unanimous votes) and clustered standard errors based on Universities and departments.

Table S13. OLS regression results for the interactive effect of URM status and joint evaluation on department negative vote percentage excluding candidates who received zero negative votes.

Variable	Department Negative Vote %	
	b	SE
URM status	.17*	.08
Joint evaluation	.03	.03
URM x Joint evaluation	-.30*	.11
Women	-.04	.03
Tenure in rank	.01*	.004
Promotion in rank	-.001	.04
H-index	-.001	.001
External grants	-.004	.002
Total department votes	-.01	.001

(7) Why not control for how many other candidates have been considered so far, in total, at the time of the candidate’s vote? Or how far into the P&T season (e.g. in days/weeks) the candidate was considered?

We agree this would be an interesting variable but unfortunately, as described above, we do not have temporal data on when these decisions were made. That said, as described in response to your Comment 2, we looked into the P&T policy at each institution in our dataset and found that in all cases they required candidates to be evaluated within a relatively short time window to meet university timelines for completing multiple levels of required reviews (roughly one month). Therefore, the P&T season for all candidates was a short time window and thus it is likely (to save time and avoid scheduling conflicts) that departments arranged for all candidates to be evaluated in the same meeting. Further, even if they are not, the ordering would be tightly arranged.

As mentioned in comment 2, we now highlight the temporal structure of the voting in a footnote in the introduction (p7), provide more details in the methods sections (p26), include the new table for reference (see Table S1), and address the limitations (and contribution) of a lack of temporal specificity in the discussion (p19).

(8) I found the justification for only looking at Black and Hispanic URMs and excluding Native Americans somewhat confusing. With an N of 2, why not include Native Americans at least in a robustness check? They are certainly URMs! The exclusion of the “Other” race category seems more justifiable since it is unclear what “Other” means.

Thank you for noting this. Our initial rationale for not including these individuals was due to the small sample and the precedent set by other research by other teams using this dataset (e.g., Masters-Waage et al., 2024). Nonetheless, we ran a robustness check including Native Americans in the analyses and found the same result, (see Table S10), and documented the replicated finding in a new section title “Robustness Checks” (see p14-15).

(9) To enhance reproducibility and transparency, I strongly encourage the authors to upload all analysis scripts, materials (for the forecasting experiment), and more detailed documentation of variable creation on their OSF.

Thank you, we now provide all analysis scripts and materials on our OSF page, the documentation of how variables were created is provided in the methods section.

Reviewer #3 (Remarks to the Author):

Authors' Response:

Thank you for your contributions to the review process.

Reviewer #4 (Remarks to the Author):

Comments on Joint Evaluation Reduces Racial Disparities in Promotion and Tenure: Evidence from a Natural Experiment

Summary:

The authors use variation in the “mode” of tenure voting, joint vs. single, across several institutions and departments using remarkable data on candidates’ attributes combined with department-by-institution information on the voting process. They begin with a survey asking whether those gatekeepers think this would improve or worsen disparities in URM tenure cases. Framed within a cross-disciplinary literature on choice architecture the authors conclude that in fact URM candidates receive fewer negative votes when evaluated in the joint, as opposed to separate, case, contrary to what the gatekeepers thought would happen.

Authors’ Response:

Thank you for the support of our dataset and the manuscript. Below we provide a point-by-point response to each of your comments.

Comments:

1. I very much appreciate framing the paper in terms of choice architecture and anchoring – in fact I think there is a little room to expand on the role anchoring plays in this particular case. I appreciated the discussion here as it sets up the mechanisms for the experiment very nicely, this was excellent framing and I appreciated that you framed the potential for it to go either way. Starting with a survey of gatekeepers’ beliefs enriched the study, in particular given the results.

Thank you for your supportive words, in the new manuscript we have taken further steps to focus the paper on choice architecture. Reading through the totality of the reviewer comments we decided to focus the paper more closely on evaluation mode from the judgment and decision making perspective.

On the encouragement of another reviewer we did decide to remove the formal hypothesis 2, i.e., that separate evaluation will improve outcomes for URM faculty. Doing so helps us by providing a stronger theoretical proposition that this paper is investigating. That said, we also agreed that highlighting that results could “go either way” - and that a significant number of forecasters believed that separate evaluation will

improve outcomes for URM faculty - demonstrates the importance of this research question. Therefore, we have moved the discussion of the competing hypothesis to the end of the forecasting study. This positions our theoretical hypothesis (that joint evaluation will improve outcomes) against the lay hypothesis of P&T decision makers (that joint evaluation will worsen or have no effect on outcomes).

For ease of reference we pasted this new section from the discussion of the forecasting study below (p8-9):

“It is particularly notable that the most common response from gatekeepers was that separate evaluation would reduce racial disparities. Although this perspective contrasts our hypothesis, it is somewhat aligned with general social psychological theories of perception, that race is more salient when individuals are in a group (2 or more) than when they are alone^{37,38}. For example, if a decision maker is evaluating a P&T candidate individually, their most prominent social feature might be that they are a faculty member; however, if they are evaluated with someone of a different race, it will become more salient that they are a Hispanic faculty member and that the other candidate is a White faculty member. In sum, the results of the forecasting survey present a competing hypothesis based on faculty lay beliefs, that separate evaluation will reduce racial disparities in the P&T process. The contrast between this lay hypothesis and our initial hypothesis drawn from behavioral decision theory sets up the natural experiment as an empirical test to resolve these two contrasting perspectives.”

2. I was unclear on how “treatment” is “assigned” here (using the terms lightly). You say in the back end of the paper that, “Joint evaluation is defined as when more than one P&T candidate is seeking promotion to the same role, in the same department, in the same academic discipline, and in the same year. Separate evaluation was any case when this was not occurring (p. 20)”. How was the mode determined? Does it vary over time within department? Does it vary across departments within a university? I think yes to both of these but the extent of this is important for understanding where the variation is coming from. Page 11 partially discusses this, but this is an important aspect to be clear on. I took time to download and see the data (thanks for providing this, it is a great practice). I did not see a year indicator in the data, so it seems as if this is all running as a pooled analysis. If you have the data by year (I assume you don’t or you would have included it) I suggest treating this as a panel.

In the revised manuscript we have provided more information on the assignment process in the introduction (p7). To answer your two specific questions: yes, it does vary over time within departments; yes, it does vary across departments within a university. Ultimately, joint evaluation is a function of when two candidates were hired in the same

year, or if not, two candidates hired in different years happen to go up together (e.g., one took an extension or went up early). Therefore, department or university has little effect on this, except that larger departments are more likely to have multiple candidates going up (which we control for).

We have pasted the new paragraph in the introduction in which we provide more details on the “assignment” process (p7):

“We examine this hypothesis in the context of P&T decisions made within US institutions. Promotion policies can vary across institutions^{34,35}, however, in general they follow a similar approach. Faculty are required to go up for promotion after a certain number of years (i.e., tenure clock), though they can go up early if they elect to and they can receive an extension if their university permits (e.g., birth of a child). Once faculty have chosen to go up for promotion, most universities’ faculty affairs units follow a rigid timeline determining, for example, when the candidate must submit their promotion portfolio and by when external review letters must be completed. The particular part of the P&T process we focus on is the meeting of the P&T committee at the department-level, to make their decision on the candidate. Department P&T committee decisions are influential since department P&T committee members are direct colleagues and usually in the same or adjacent disciplines as the P&T candidate.

In this paper, we define joint evaluation as when more than one candidate is being evaluated within this allotted window. In other words, there is more than one P&T candidate going up for promotion to the same rank, in the same department, in the same promotion cycle. Notably, this is a less rigid designation of joint evaluation compared to past laboratory studies where joint evaluation occurred at exactly the same time and was a forced-choice between two options²⁶, that said, the proximity of the evaluations still suggests that decision makers will use the candidates as a reference point for each other^{27,36}.”

As to whether we have the data by year, we do. We have also taken your advice and conducted a robustness check treating this data as a panel. To do so, we transposed the current dataset into a panel data with departments within each University as ID and year as the time variable. The panel data includes 326 unique departments in 7 years. We report these analyses in response to your more specific comment (C4) below concerning the time element.

3. Tables 1 and 2 could be expanded to provide a clearer picture of the populations. I suggest having table 1 as only a table of means (and SD’s) of the universe of schools (or departments). I will assume that the pool is balanced (each school is in the data in each

year). We might have statistics about the universities (size, category, possibly some measure of “rank” or competitiveness) separately by single vs. joint evaluations. I understand that there are 10 universities but later you say four universities had unusable data later on in the paper.

Thank you for this suggestion, we have created a new table that provides the means (and SDs) for each of the study variables broken down by school. Given that the primary analyses are conducted collapsing across schools, we decided to keep Table 1 in the main manuscript and add this new table in a supplement (see Table S2).

On the request of another reviewer we have also expanded Table 2 to include a breakdown by URM status (see Table 2)

Due to the sensitivity of the data we intend to keep the identity of the universities as concealed as possible, therefore, we are hesitant to include detailed statistics about the universities. However, we have included a new section of the methods in which we go into more general details about the universities in our sample (p25-26; see below).

“Our team built a consortium of universities to study promotion and tenure processes. In building this consortium, the goal was to bring together a diverse set of institutions in terms of size, research status, and ranking. In terms of student size the consortium includes two smaller institutions (5,000-15,000 students), three medium-sized institutions (40,000-50,000), and one larger institution (60,000+ students). The consortium primarily includes universities classified as Carnegie R1 institutions (i.e., very high research activity) but also one university classified as Carnegie R2 institutions (i.e., high research activity), providing variation of research activity. Further, whilst all institutions rank in the top 500 universities in the US according to the US news ranking, their position varies from 40th to 450th with the median position being 157th. In addition, the set of universities includes two minority serving institutions, one designated as a Hispanic-Serving Institution (HSI) and another as a Historically Black College or University (HBCU). Finally, the six institutions span four US states providing geographical diversity.

All institutions were allotted approximately one month for P&T decisions to be made at the department level. Therefore, when multiple candidates were evaluated in the same year they were both considered within this time window. For a full breakdown of the P&T timeline at each institution see Table S1; note, the identity of the university was masked to preserve anonymity.”

Also, thank you for allowing us to clarify about the institutions in our data vs. those in the consortium. Initially, the consortium that came together to create this dataset included 10 institutions. However, for a variety of reasons not all of the data is available for analysis. First, two institutions have not completed the coding process, so we cannot include their data. Second, after construction of the consortium we learned that two of the smaller institutions had non-traditional ways of completing P&T voting, e.g., they did not clearly structure P&T committees or record votes. So, we could not include this data in the analyses. Therefore, these four institutions are not included in analyses as none of them have data on voting outcomes; the focal dependent variable. We now explain this more clearly in a footnote (see p9).

4. In the main model, you have institution and CIP code controls, meaning that you have what I believe are university and separately department (or field) fixed effects (from the table notes). This means that the variation could be coming from changes within university-departments over time, or is coming simply off of differences across departments within universities, possibly over time or not, it is unclear which, but is important. The data available for download show that there is within department variation, implying that either there were changes over time or the variation is simply random across years, most likely the former. One would have to assume that the switch across years was uncorrelated with any other changes that are related to the vote share, which is possible but unlikely. I might also suggest a first regression where the dependent variable is the treatment itself – joint eval, such that the reader can understand which universities/departments are more likely to feature this mode. Without a time element however, you can only learn so much from this.

We used a panel data analysis approach to verify the results. The panel data consisted with 7 unique years’ P&T decisions: ranging from 2016 - 2022 and 326 unique departments. First, we examined whether there is significant difference in joint evaluation across Universities and departments. The results showed that there was no significant difference in joint evaluation across Universities ($F(5,1798) = 2.15, p = .06$). The Tukey post-hoc analysis showed that none of the pairwise differences between schools are statistically significant at the 0.05 level.

In addition, we replicated the analyses using a panel data approach. We first conducted a Hausman test to determine the appropriate analysis to perform. The results of the Hausman test suggest using the random effect model ($\chi^2 = 4.59, df = 3, p = .20$). The panel data results are shown in Table S14.

Table S14. Stepwise random effect models for the interactive effect of URM status and joint evaluation (also termed Joint Eval.) on department negative vote percentage.

Variable	Department Negative Vote %
----------	----------------------------

	No Controls		Base Controls		Productivity		Department Size		All Controls	
	b	SE	b	SE	b	SE	b	SE	b	SE
URM status	.10***	.02	.09***	.02	.07*	.03	.09***	.02	.07*	.03
Joint evaluation	.04***	.01	.04***	.01	.03**	.01	.04***	.01	.03**	.01
URM x Joint eval.	-.10**	.03	-.09**	.03	-.13**	.04	-.09**	.03	-.13**	.04
Women			-.003	.01	-.01	.01	-.004	.01	-.004	.01
Promotion rank			.02*	.01	.04**	.01	.02	.01	.04*	.01
Tenure in rank			.01**	.002	.01***	.002	.01**	.002	.01***	.002
H-index					-.001	.001			-.001	.001
External grants					-.00	.00			.00	.001
Total department votes							-.00	.001	-.00	.001
R ²		.02		.07		.08		.07		.08

5. I believe you're using a linear probability model (i.e. OLS) which is probably just fine, but because the dependent variable is a percentage (0-1) and is therefore bounded, using a generalized linear model (GLM) with a logit-link might improve precision. I also suggest clustering standard errors on universities (or departments), given the marginal results I think it's important to take inference seriously here. I tried this with the data and code provided and the standard errors shrink, which is surprising, but shows there is more precision to be had. Likewise, given there is variation within university-CIP cells in joint evaluation you can actually have a university-by-CIP fixed effects and results are similar.

We appreciate the suggestions and have followed your advice.

We first conducted the GLM analysis with logit-link by coding 1 = any negative vote, 0 = unanimous for the logit model. The results were shown in Table S6 below. The result patterns were consistent with those of OLS model with the p value of the interaction shrinking to .08. Note, in the paper we found that there was no interactive effect on unanimous votes so this result is consistent. This result is reported in the manuscript on p13.

Table S6. Generalized Linear Modeling analysis with logit-link for the interactive effect of URM status and joint evaluation (also termed Joint Eval.) on department vote outcome (binary: 1 “any negative vote”, 0 “no negative votes”)

Department Negative Vote (1 = any negative vote, 0 = unanimous)		
	b	SE
URM status	.43	.37
Joint evaluation	.37*	.17
URM x Joint eval.	-.95	.55
Women	.03	.17
Promotion rank	.82	.19
Tenure in rank	.04*	.02
H-index	-.001	.007
External grants	.01	.01
Total department votes	.07****	.001
McFadden R ²	.41	
Cox and Snell R ²	.50	

We also clustered the standard error on University and department to check the robustness of the results. The results are shown in Table S8 and reported in the paper on p14-15.

Table S8. OLS regression models for the interactive effect of URM status and joint evaluation (also termed Joint Eval.) on department negative vote percentage using clustered standard errors for universities (Model 1) and departments (Model 2).

Cluster SE on Universities		Cluster SE on Departments	
b	SE	b	SE

URM status	.07	.05	.07	.04
Joint evaluation	.03***	.003	.03	.01
URM x Joint eval.	-.12*	.04	-.12*	.05
Women	-.01	.01	-.01	.01
Promotion rank	.04***	.01	.04**	.01
Tenure in rank	.01***	.001	.01*	.002
H-index	-.00001	.001	-.00001	.001
External grants	.001	.0004	.001	.001
Total department votes	.0004	.0003	.0004	.004
R^2		.08		.08

Lastly, we ran the university-by-CIP fixed effects. We identified 103 University-by-CIP cells and ran the fixed effect using fixest package (see Table S9). The results are consistent with OLS results. We report these results on p14-15 in the manuscript.

Table S9. OLS regression model for the interactive effect of URM status and joint evaluation (also termed Joint Eval.) on department negative vote percentage using university-by-CIP fixed effects

	Department Negative Vote %	
	b	SE
URM status	.06**	.02

Joint evaluation	.02*	.01
URM x Joint eval.	-.12**	.03
Women	-.0004	.001
Promotion rank	.04**	.01
Tenure in rank	.006***	.001
H-index	.0002	.001
External grants	-.001	.001
Total department votes	.001	.001
R^2		.06

6. I'd also suggest putting the sample sizes in the results table. You note some drop out due to missing data – you could remedy that by setting the covariates for those observations (assuming they're not the dependent variable) to zero and including an additional dummy variable indicating that the control was missing to get some sample size back and rule out any issues from attrition across models. You lose about half the sample from the first to the last regression. Unless that attrition is random, it is worth figuring out which variables are causing the issue – it's difficult to determine how much the sample change affects inference. At the very least the reader should be aware of this.

Below we provided a missing data summary. There is significant missing data in controls such as h index. This is because h index was retrieved from faculty google scholar pages, however, not all faculty in the data set had google scholars, meaning that their data was missing.

The presence of missing data is a major reason why we performed our analyses in a stepwise fashion, demonstrating that the results replicate when adding more controls and

by virtue reducing the sample size. In the manuscript we now make this decision more explicit and highlight to the reader the decreasing sample size by indicating the N for each model and include the N for each model in the tables.

Below for your reference is a Table S3 which provides a count of the missing data.

Table S3. Count of missing data for each variable used in analyses.

Variable	# Missing
Joint Evaluation	0
Candidate Gender	0
Number of Department Votes	0
URM Status	58
Grants Awarded as PI	196
Department Negative Vote Percentage	209
Candidate H Index	573

We also conducted a missing indicator approach by creating dummy variables for external grants and h index with missing values. The results are shown in Table S11 below and mentioned on p15.

Table S11. Interactive effect of URM status and joint evaluation (also termed Joint Eval.) on department negative vote percentage with missing value dummy variables.

	Department Negative Vote %	
	b	SE
URM status	.09***	.02

Joint evaluation	.04***	.01
URM x Joint eval.	-.10**	.03
Women	-.01	.01
Promotion rank	.02	.01
Tenure in rank	.004*	.001
H-index	.0004	.001
H-index_missing	.01	.01
External grants	-.001	.001
External grants_missing	.01	.02
Total department votes	-.001	.001

7. As an even smaller note, I'd suggest making the dependent variable positive votes as opposed to negative, it's just easier to interpret the effect of joint evaluation as a "positive" effect. I apologize for what might seem like nit-picking here, please do not hear me saying I don't believe (or like) the results, the opposite is true – I want to make sure they are as strong, reliable, and clear as possible. Now on to actual comments.

Thank you for this comment, certainly not nit-picking as it is a question we have thought about a lot as a team. We agree referring to positive vote percentage could make some sentences in this paper more intuitive to understand. However, given that the majority of votes at the department level are positive and many candidates can receive positive unanimous votes, we focused on negative votes as they are a deviation from the modal response. Further, just one negative vote can have a significant effect as it makes a vote non-unanimous.

Beyond this rationale, we are also aware that this dataset has been worked on previously by other scholars (and planned for future scholars) and the standard has been set that researchers refer to "negative vote percentage" as opposed to "positive vote percentage".

Therefore, while this decision is somewhat arbitrary, to provide consistency across projects we have decided to keep “negative vote percentage” as the focal variable.

8. The dependent variable has a pretty limited distribution. The data suggest that more than three-quarters of the votes were unanimous, meaning there is very little variation in the dependent variable to begin with. Possibly this explains the very small R2, and why the unanimous variable outcome does not show effects. This is pretty important for the analysis, and for the reader.

Thanks for suggesting this. The table below shows the distribution of the department negative votes. It shows that 75.5% of the votes were unanimous.

Distribution of the Department negative vote %

Data category	Percentage
0	75.5%
0-0.1	12.3%
0.1-0.2	6.5%
0.2-0.5	4.0%
0.5-1	1.7%

Thank you for the suggestion that the relatively small variation in the dependent variable may explain the small R2 as this appears to be the case. We ran the same model with a subsample without unanimous votes, including 375 candidates. The results were consistent with these of the whole sample and R2 increased to around 0.27. These results are reported on p15 of the manuscript.

Table S13. OLS regression results for the interactive effect of URM status and joint evaluation on department negative vote percentage excluding candidates who received zero negative votes.

Variable	Department Negative Vote %	
	b	SE
URM status	.17*	.08

Joint evaluation	.03	.03
URM x Joint evaluation	-.30*	.11
Women	-.04	.03
Tenure in rank	.01*	.004
Promotion in rank	-.001	.04
H-index	-.001	.001
External grants	-.004	.002
Total department votes	-.01	.001
R^2	.27	

N = 375.

9. I may be reading the table wrong, but my interpretation is that on net, URM candidates receive the same share of votes. The main effect (e.g. Model 1 of Table 3) has a positive coefficient of 0.10 for URM's and the interaction effect is -0.09. I take this to mean that in that model (1) in joint evaluations, URM's receive the same share of negative votes but when evaluated separately, they receive 9 percentage points fewer votes. I believe percentage points is the right interpretation here as the dependent variable is the percent of votes. I think you could add the baseline negative vote share to the table and then calculate what 0.09 is off of the base share of negative votes. On line 266 and in the abstract you say that the result is a 9% reduction in negative votes, which I believe is incorrect.

Thank you for providing us with the opportunity to clarify. What is agreeably a bit confusing is that the positive coefficient of 0.10 means that URM faculty are receiving more negative votes. The interaction term being negative indicates that this decreases in joint evaluation. The 9% number is coming for the generation of the simple slopes (see p13), which indicate that for URM faculty specifically that the relationship between evaluation mode and department negative vote percentage has a coefficient of 0.09.

To help make this clearer we have taken a number of steps. First we provide a model without the interaction (see Table 3), to indicate that URM status is associated with a higher negative vote percentage overall.

Second, we provide more description in the abstract to indicate where the 9% number comes, we now state, *“In joint evaluation, analysis of the simple slopes, found that URM (Black and Hispanic) faculty received (on average) 9% fewer negative votes than in separate evaluation.”* We hope this makes things clearer to the reader from the offset.

Finally, on the suggestion of another reviewer, we have also conducted analyses to conceptualize the downstream effect of a shift in negative vote percentage on the Provost vote. At the institutions in this dataset the Provost vote represents the ultimate decisions on whether a candidate will receive a promotion (1) or not (0). Therefore, to contextualize our result, we calculated the conditional indirect effect of URM status on provost vote via department-level negative vote perception, depending on whether candidates were evaluated jointly or separately. In other words, how did the benefits of joint evaluation for URM faculty at the department level translate into likelihood of receiving a positive provost vote. Moderated mediation analyses found support for a conditional indirect effect (Index of moderated mediation = .65, 95% bootCI = [.18, 1.25]); see Table S7), meaning that the interaction between URM status and evaluation mode had downstream effects on provost vote via department level voting. Post-hoc tests found that, for a URM faculty, being evaluated jointly lead to a 16.2% increase in the likelihood of receiving a positive provost vote. We provide these analyses in the results section (p14) and mention the result in the abstract.

10. I was also struck by the fact that H-index and grants have no relationship to vote shares, and by the lack of negative effect for female candidates. Do these fit with other results from the literature? You might consider interacting these with URM (and with joint), but I am not clear on what the story to tell would be. To add, the R2 is quite low considering you have CIP and university FE in the model (unless you reported the adjusted R2, which is fine but just say so). It’s hard to believe that all of those explain only a tiny fraction of the overall variance is explained by all of that information. Possibly this is due to the lack of variation in the dependent variable.

Thank you for noting this, we agree it is an interesting point and now pick up on it in the discussion (p22-23). We think this speaks to the intense complexity of P&T decisions, and the importance of a factor such as evaluation mode over specific metrics.

Interestingly, past research has shown that factors like h index may have an indirect effect on P&T outcomes via content of external review letters (Madera et al. (2024) External review letters in academic promotion and tenure decisions are reflective of reviewer characteristics. *Research Policy*, 53(2), 104939) and influencing perceptions of minority candidates (Masters-Waage et al. (2024). Underrepresented minority faculty in the USA face a double standard in promotion and tenure decisions. *Nature Human Behaviour*, 8(11), 2107-2118) but it appears it is not a direct predictor of voting outcome.

Nevertheless, given the relevance of research output in the P&T process we still think it is appropriate to include these controls.

For ease of reference provide the added section to the general discussion below (p22-23):

“Another challenge associated with the use of a real world dataset is the complex inputs to decisions. As shown in Table 3, the model including all of the control variables accounts for 8% of the variance in negative vote percentage at the department level. An r -squared below 10% is not uncommon in applied psychological science^{59,60}, given the inherent complexity and noise in the psychological process⁶¹. Further, as shown in the robustness checks, the r -squared increases to 27% when addressing skew in the dataset by excluding participants with no negative votes. Nevertheless, it does highlight the challenges in predicting P&T voting behavior. This is notable given that measures of scholar productivity (h index and grants obtained) are included in the model and neither of them significantly predicted voting outcomes. Although past research has identified indirect effects of scholarly productivity on voting outcome^{2,62} the lack of a direct relationship demonstrates that P&T decision makers derive evaluations of research impact and quality through means other than citation metrics, such as relying more on their qualitative assessment of the candidates’ contributions rather than on cumulative metrics⁶⁶. Instead, it appears that P&T decision making is a more complex process, highlighting the need for more research into this topic in order to identify the factors driving these high-stakes decisions. ”

Further, in line with our response to Comment 8 above, we also think the lack of variation in the dependent variable also plays a role.

11. I think you can learn something from the vote share vs. the unanimous vote vis-a-vis your hypotheses, in particular the anchoring effect. That it is not statistically significant does not imply the result is not meaningful. Nonetheless, I think Figure 2 is worth highlighting more - non-URM votes are unaltered by the switch, but negative URM votes decline by two-thirds (10 percentage points off of a base of 15 percent)! You could do this for unanimous vs. not as well. I think this says something about the mechanism – opinions about non-URM’s don’t change, URM’s are simply seen in a less negative light, which I think points more to the double-standard mechanism you mentioned in the framing section more than the negative stereotyping, though it is tough to pin the mechanism down.

Thank you for this suggestion. We did look into the effect for unanimous votes and found the effect not to be significant, we now report this figure in a supplement. We include a new subsection of the results focusing specifically on unanimous votes.

More critically, we agree that this finding does provide support inline with the double standard mechanism. In essence the double standard hypothesis is somewhat related to stereotyping, as it is one's prior beliefs about a group (i.e., stereotypes) that lead to the double standard. That said, your comment encouraged us to give more attention to the double-standard mechanisms in the introduction and the discussion. This also provided us with the opportunity to discuss figure 2 in more depth (p19). Note, due to the updates in the analysis in response to the reviewers the effect for non-URM faculty is now significant with these individuals being evaluated more harshly in joint evaluation, receiving 3% more negative votes. We now mention this in the results and pick up on it in the discussion (see p19).

Finally, we would like to thank you for your thorough review and hope our comments have sufficiently addressed them.

REVIEWER COMMENTS

Reviewer #1 (Remarks to the Author):

The authors have adequately addressed my concerns.

Thank you for your support in the review process.

Reviewer #2 (Remarks to the Author):

I thank the authors for their thoughtful and comprehensive response to the reviewers' comments. I appreciate the point-by-point responses provided in their rebuttal letter.

The revised manuscript shows improvements in several areas. Most notably, the authors have streamlined the theoretical framing by removing the competing Hypothesis 2. I also appreciate the detailed information about the tenure process, the expanded analyses (including gender effects and various robustness checks), and the broader positioning of the findings beyond academia.

I believe that the current version of this manuscript is very strong and I only have a few minor comments at this point.

Thank you for your supportive words. We addressed the additional comments, which helped significantly streamline and strengthen the paper. All suggestions were addressed through contained changes in the manuscript. We provide a detailed response to each comment (in **bold**) indented below (in plain text).

1. I appreciate the authors' effort to assess practical significance by linking evaluation mode to provost outcomes through negative vote share using a moderated mediation model. However, this approach restricts the analysis to a single pathway and assumes the effect of evaluation mode on the provost's decision operates only through department-level votes. While this is useful for isolating that channel, I recommend also presenting the total effect of evaluation mode on the provost decision (e.g., regressing the provost vote directly on the URM x Joint Evaluation interaction). This would provide a complementary and more policy-relevant estimate of the intervention's overall impact.

We appreciate your point here and the policy-relevance of the conditional main effect. We did not conduct this analysis previously as it does not directly test our theory of whether joint evaluation affects evaluation of URM faculty. This is because at the provost level there is almost always joint evaluation, as a provost will see numerous P&T candidate portfolios every year. Therefore, our effect is theoretically confined to the department-level where the evaluation mode varies.

That said, from a practical perspective we do see the value of estimating the conditional direct effect as this demonstrates the rates of promotion for candidates in each evaluation mode. Therefore, we have included this estimate in the manuscript (see p15):

"We also examined the conditional main effect of URM status, i.e., regressing provost vote on the interaction between URM status and evaluation mode. The interaction term from these analyses was in the expected direction but was not statistically significant ($b = -.03$, $SE = .04$, $p = .46$)."

2. I did not find Figure 2 to be a terribly compelling visual display of the key interaction effect and might suggest considering other ways of presenting the effect visually that would be more intuitive to general audiences, perhaps with a simple bar chart. It would also be helpful to state in the Figure 2 notes which regression estimates were used to generate the image presented.

We experimented with different visualizations and are grateful for your suggestion as we decided to go with a bar chart. It is indeed a better graphical representation of results in the manuscript, as shown in the figure below:

Figure 2. Relationship between URM status and negative vote percentage at the department level based on regression estimates from interaction effect in the base model (i.e., no control variables).

3. Ideally Table 2 would not only show summary statistics describing faculty and their departments by joint versus separate evaluation but would also present t-tests or two sample proportion tests to assess the balance of these summary statistics across conditions to provide statistical support for the claim that this was a valid quasi-experiment (reporting results in the section entitled “Quasi-Experimental Design” from page 9-12). An overall F-test could be used to assess balance given that some individual covariates are likely to be slightly imbalanced across the many tests conducted.

We have revised Table 2 to include t-tests for each comparison and reported these results in the section titled “Quasi-Experimental Design”, see p11. We ran separate t-tests as we did expect one variable to differ across the conditions. That was total department votes, as it is expected that larger departments will be more likely to have more joint evaluations occur.

However, as shown, this is the only variable that significantly differed across conditions and is included as a control variable. With this addition of a row, we also separated the URM status comparison into a new separate table (Table 3). We provide Table 2 below:

Table 2. Breakdown of the demographics for separate vs. joint evaluation. The final row reports Welch’s t-test for comparison across conditions for each variable.

	Women	H-Index	Extensions	Total Department Votes	Years in Present Rank	US News University Ranking	External Grants as PI	URM Status
Separate evaluation	38.20%	14.96	7.35%	8.72	6.06	138.49	4.84	12.74%
Joint evaluation	36.09%	15.80	7.06%	10.35	6.06	133.96	5.16	11.31%
T-test (t)	0.92	-1.43	0.57	-3.92***	0.02	1.46	-0.85	0.92

* $p < .05$; ** $p < .01$; *** $p < .001$

4. In my first read of the manuscript, I had not picked up on the fact that these votes took place at universities that are not, for the most part, terribly prestigious. I think providing this context in the abstract and introduction (that this analysis was done at primarily R1 universities ranked outside the top 100 in the United States) would be helpful. The results could plausibly differ at highly selective institutions, so being explicit about where this work was done up front seems important.

We have included a statement highlighting this in the introduction. We do think it is a relevant point and mention in the context of the breadth of our sample which includes a range of universities including public and private, and large, medium, and smaller institutions. In developing our consortium partnership, we intentionally pursued the inclusion of a broad set of institutions that represent Higher Education in the U.S. more broadly than a smaller, more prestigious set of universities. Having that said, while our dataset does not include top 10, highly prestigious institutions, the dataset does contain universities ranked in the top 50 and top 100. We think this variety helps strengthen our conclusions. Below is the revised sentence from the introduction stating this (see p7):

“We examine this hypothesis in the context of P&T decisions made within a diverse set of six US public and private institutions, varying in size (~10,000 to ~80,000 students), from across four states, and ranging in their US News national university ranking (rounded to the nearest ten universities rank as 50th, 50th, 130th, 170th, 170th, and 390th in the nation)”

Reviewer #3 (Remarks to the Author):

Thank you for your support for the next generation of scientist-reviewers!